# Active Sequential Posterior Estimation
# for Sample-Efficient Simulation-Based Inference

**Sam Griesemer**[1]    **Defu Cao**[1]    **Zijun Cui**[1,2†]    **Carolina Osorio**[3,4]    **Yan Liu**[1]

[1]USC    [2]MSU    [3]Google Research    [4]HEC Montréal

{samgriesemer,defucao,yanliu.cs}@usc.edu

cuizijun@msu.edu

osorioc@google.com

## Abstract

Computer simulations have long presented the exciting possibility of scientific insight into complex real-world processes. Despite the power of modern computing, however, it remains challenging to systematically perform inference under simulation models. This has led to the rise of simulation-based inference (SBI), a class of machine learning-enabled techniques for approaching inverse problems with stochastic simulators. Many such methods, however, require large numbers of simulation samples and face difficulty scaling to high-dimensional settings, often making inference prohibitive under resource-intensive simulators. To mitigate these drawbacks, we introduce active sequential neural posterior estimation (ASNPE). ASNPE brings an active learning scheme into the inference loop to estimate the utility of simulation parameter candidates to the underlying probabilistic model. The proposed acquisition scheme is easily integrated into existing posterior estimation pipelines, allowing for improved sample efficiency with low computational overhead. We further demonstrate the effectiveness of the proposed method in the travel demand calibration setting, a high-dimensional inverse problem commonly requiring computationally expensive traffic simulators. Our method outperforms well-tuned benchmarks and state-of-the-art posterior estimation methods on a large-scale real-world traffic network, as well as demonstrates a performance advantage over non-active counterparts on a suite of SBI benchmark environments.

## 1   Introduction

High-fidelity computer simulations have been embraced across countless scientific domains, furthering the ability to understand and predict behaviour in complex real-world systems. Modern computing architectures and flexible programming paradigms have further lowered the barrier to capturing approximate models for scientific study in silico, enabling wide-spread use of computational experiments across disciplines. However, despite the relative ease of capturing real-world generative processes programmatically, the resulting black-box programs are often difficult to leverage for inverse problems. This is a common challenge in practical applications; the simulator is often computationally expensive to evaluate, its implicit likelihood function is generally intractable, and the dimensionality of high-fidelity outputs is typically prohibitive. To address these issues, likelihood-free inference methods have been introduced, operating under the broadly applicable assumption that no tractable likelihood function is available. Early success along this direction was achieved through easy-to-use methods like Approximate Bayesian Computation (ABC) [41, 6], or extensions of kernel density estimation. The scale of real-world applications demands more flexible and scalable approaches, which has lead to the integration of aptly suited deep learning methods in likelihood-free

---

[†]Work completed while at USC.

38th Conference on Neural Information Processing Systems (NeurIPS 2024).

settings. Neural network-based methods [33, 29, 17] have since been proposed, introducing greater flexibility when approximating probabilistic components (e.g., the posterior, likelihood ratio, etc) in the inference pipeline. The use of the term "simulation-based inference" (SBI) has since been colloquially embraced [11] when referring to this emerging class of techniques.

SBI methods primarily leverage deep learning through their use of neural density estimators (NDE), neural network-based parametrizations of probability density functions. Common choices of NDE in practice include mixture density networks [8] and normalizing flows [40, 34], along with popular extensions (e.g., Real NVP [12], MAE [16], MAF [35], etc). Methods also vary in the probabilistic form they elect to approximate; the posterior [33], the likelihood [36], and the likelihood ratio [19, 13] are all common choices for well-established methods.

While many SBI techniques leverage basic principles from active learning (AL), they are mostly established as a helpful heuristic for increased sample efficiency, rather than an explicit optimization over a defined acquisition function. For example, methods like Sequential Neural Posterior Estimation (SNPE) [33] boost sample efficiency (over non-sequential methods) by iteratively updating a proposal prior $\tilde{p}(\theta)$, steering simulator parameters to values expected to be more useful for learning the posterior under observations $x_o$ of interest.

While sequential proposal updates are an effective first-order step to more informative simulation runs, there are many key factors that remain overlooked. For example, the updated proposal does not take into account the current parameters of the NDE itself, and parameter samples in the batch are drawn from the current proposal independently. This fails to fully utilize myopic AL strategies and batch optimization, leading to large amounts of simulation runs with high expected information overlap and wasted computation. To address this issue, we formulate an active learning scheme that (1) selects samples expected to target epistemic uncertainty in the underlying probabilistic model, and (2) makes acquisition evaluation simple and efficient when using any Bayesian NDE.

We demonstrate the effectiveness of our method on the origin-destination (OD) calibration task. OD calibration aims to identify OD matrices that yield simulated traffic metrics that accurately reflect field-observed traffic conditions. It can be seen as a parameter tuning process, akin to model fitting in machine learning. From the machine learning perspective, OD calibration presents challenges due to the requirement of calibrating specific unique samples from observed traffic information, such as link flows, trip speeds, etc.

Our contributions are summarized as follows:

- Active Sequential Neural Posterior Estimation (ASNPE), an SNPE variant that incorporates active acquisition of informative simulation parameters $\theta$ to the underlying (direct) posterior estimation model, without the use of additional surrogate models. This helps to drive down uncertainty in parameters of the utilized NDE and improve sample efficiency, both of which are particularly important when interfacing with computationally costly simulation-based models.

- An efficient approximation to the proposed acquisition function above, along with a means of training Bayesian flow-based generative models for density estimation during posterior approximation (both with open source implementations[3]). Leveraging this class of models enables direct uncertainty quantification in the acquisition function, and is more flexible, efficient to train, and scalable to high-dimensional data than many traditional Bayesian model choices (e.g., Gaussian processes).

- A Bayesian formulation of the OD calibration problem and coupled statistical framework for performing sequential likelihood-free inference with neural posterior estimation methods. We show ASNPE outperforms baseline methods across a wide variety of simulation scenarios on a large-scale traffic network. We also evaluate ASNPE on three broader SBI benchmark environments and find it acheives a performance advantage over non-active counterparts.

---

[3] https://github.com/samgriesemer/seqinf

## 2 Background

### 2.1 Neural posterior estimation

Given observational data of interest $x_o$ and a prior $p(\theta)$, we want to carry out statistical inference to approximate the posterior $p(\theta|x = x_o)$ under the model $p(x|\theta)$. We assume $p(x|\theta)$ is defined implicitly via a simulation-based model, where direct evaluation of $p(x|\theta)$ is not possible but samples $x \sim p(x|\theta)$ can be drawn. Conventional Bayesian inference is thus not accessible in this setting, and we instead look to approximate the posterior using $N$ generated pairs $\{(\theta_i, x_i)\}_{i=1}^N$.

Neural Posterior Estimation (NPE) methods attempt to approximate the posterior directly with a neural density estimator $q_\phi(\theta|x)$ trained on samples $\{(\theta_i, x_i)\}_{i=1}^N$, where $\theta_i \sim p(\theta)$ and $x_i \sim p(x|\theta_i)$, by minimizing the loss

$$\mathcal{L}(\phi) = \mathbb{E}_{\theta \sim p(\theta)} \mathbb{E}_{x \sim p(x|\theta)} \left[ -\log q_\phi(\theta|x) \right]$$

for learnable parameters $\phi$. Provided $q_\phi$ is sufficiently expressive, $q_\phi(\theta|x)$ will converge to the true posterior $p(\theta|x)$ as $N \to \infty$.

Sequential Neural Posterior Estimation (SNPE) methods break up the NPE process across several iterations, and can improve sample efficiency by leveraging the fact that $p(\theta|x = x_o)$ is often far more narrow than $p(\theta)$. While accurately representing $p(\theta|x)$ for any $x \in \mathcal{X}$ is ideal (where $\mathcal{X}$ is the simulation output space), doing so can require prohibitively large simulation samples, including outputs from parameters with low posterior density under $x_o$. To combat this, SNPE methods draw $\theta$ expected to be more informative about $p(\theta|x_o)$ by using a successively updated proposal distribution $\tilde{p}(\theta)$ which approximates $p(\theta|x = x_o)$. Training the NDE $q_\phi$ on samples $\tilde{\theta} \sim \tilde{p}(\theta)$ when $\tilde{p}$ is not the true prior, however, will cause it to converge instead to the proposal posterior $\tilde{p}(\theta|x) = p(\theta|x) \frac{\tilde{p}(\theta)p(x)}{\tilde{p}(x)p(\theta)}$, rather than the true posterior (as shown in [33]). Existing SNPE methods correct for this in different ways: *SNPE-A* [33] trains $q_\phi(\theta|x)$ to approximate $\tilde{p}(\theta|x)$ during each round and employs importance reweighting afterward, *SNPE-B* [29] directly minimizes an importance weighted loss, and *SNPE-C* [17] (also known as Automatic Posterior Transformation, or APT) maximizes an estimated proposal posterior that easily transforms to the true posterior.

### 2.2 Bayesian active learning

Bayesian active learning is a selective data-labeling technique commonly employed in data-scarce learning environments. Active learning assesses the strength of candidate data points using a so-called acquisition function, often capturing some notion of expected utility to the underlying model given the currently available data. Given an acquisition function $\alpha$, computing the next best point to label includes optimization of $\alpha$ over a domain of as yet unlabeled points $U$: $x^* = \text{argmax}_{x \in U} \alpha(x, p(\theta|D))$, where $x$ are input data points and $p(\theta|D)$ is the posterior of the Bayesian model parameters $\theta$ given the current training dataset, i.e., the distribution over parameters after training the model. In modern Bayesian deep learning pipelines, this model is often a Bayesian neural network [15, 21]. Many acquisition functions used in practice are extensions or approximations of expected information gain (EIG) [20]

$$\text{argmax}_x \mathbb{H}(\theta|D) - \mathbb{E}_{y \sim p(y|x,D)} \Big[ \mathbb{H}[\theta|D \cup \{(x, y)\}] \Big], \tag{1}$$

where $\mathbb{H}[\cdot|\cdot]$ is conditional entropy, and $y$ are input labels.

Several existing works explore the use of Bayesian optimization (BO) in the likelihood-free inference setting. [18, 22] employ Gaussian processes (GPs) as surrogate models for the discrepancy as a function of $\theta$, and select parameter candidates by optimizing this surrogate with BO. GPs are also used as a surrogate by [30] to represent the proposal distribution in MCMC ABC. [4] further extends these principles to deep Gaussian processes and leverage these models as surrogate likelihoods. In this work, we explicitly avoid the use of likelihood surrogates and aim to leverage only the approximate posterior NDE model, with the express intent of subverting additional computational overhead and enabling the use of powerful NDEs (e.g., flow-based generative models).

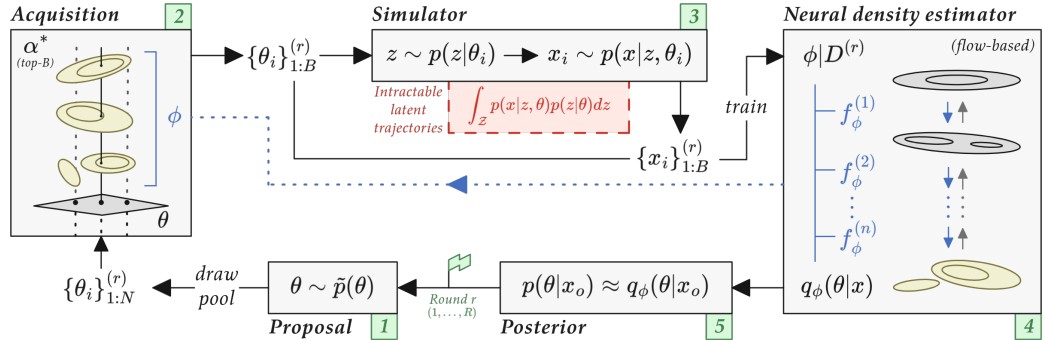

Figure 1: **Depiction of the proposed active learning-integrated method.** Demonstrates the high-level ASNPE pipeline. Samples $\theta_i$ are drawn from sequentially updated proposal distributions $\tilde{p}(\theta)$, filtered according to the acquisition function $\alpha(\theta_{1:N}, p(\phi|D))$, and run through the simulator $p(x|\theta)$ to generate $B$ pairs $(\theta_i, x_i)$ for training the approximate posterior $q_\phi$. The learned posterior is then conditioned by the target observation $x_o$, producing the next round's proposal $\tilde{p}(\theta) = q_\phi(\theta|x_o)$.

## 2.3 Origin-destination calibration

OD calibration is an important task for transportation agencies and practitioners who develop traffic simulation models of road networks and use them to inform a variety of planning and operational decisions. Calibrating the input parameters of these simulators is an important offline optimization problem that agencies must face on a regular basis (e.g., when new traffic data are made available, when changes to the road network have occurred, etc). These simulators are often computationally expensive to evaluate, however, and highlight the practical importance of developing sample efficient calibration methods.

Existing works primarily approach OD calibration using general-purpose simulation-based optimization (SO) algorithms, such as Simultaneous Perturbation Stochastic Approximation (SPSA) methods [5, 47, 25, 7], genetic algorithms [24, 43, 47], and neural interaction models [48]. Many general-purpose SO algorithms tend to require large numbers of simulation evaluations, which can be computationally costly. To address this issue, recent extensions of SPSA have been proposed [10, 27, 1, 45]. Analytic metamodels have also been considered and shown to reduce the need for large numbers of function evaluations [31, 32, 49, 3].

## 3 Methodology

### 3.1 Active learning for SNPE

SNPE methods produce iteratively updated proposal priors $\tilde{p}^{(r+1)}(\theta) \approx p(\theta|x = x_o)$ from $q_\phi(\theta|x_o, D^{(r)})$ at each round $r$ in the inference process, where $D^{(r)}$ is the accumulated dataset $\{(\theta_i, x_i)\}_{i=1}^{rB}$ by round $r$ and $B$ is the number of newly collected pairs per round. These sequentially updated proposals provide a means of drawing parameter values $\theta^{(r)} \sim \tilde{p}^{(r)}(\theta)$ that are increasingly useful (i.e., high likelihood) to the posterior estimate of interest $p(\theta|x_o)$. As such, SNPE offers a clear possible benefit for improved sample efficiency over non-sequential NPE, which cannot explicitly sample new values at expected high likelihood regions under $x_o$. Despite this, it's difficult to quantify theh value of any particular $\theta \sim \tilde{p}^{(r+1)}(\theta)$ and whether it's worth the computational cost to obtain $x \sim p(x|\theta)$ with respect to its utility to the underlying NDE $q_\phi(\theta|x, D^{(r)})$.

In high-cost simulation environments, we want to take every measure to sample only at highly informative regions of the parameter space to improve our estimate $q_\phi(\theta|x_o)$ of $p(\theta|x = x_o)$. This entails a more principled analysis of candidate simulation parameters $\theta$ before investing in the simulation run $x \sim p(x|\theta)$. In an attempt to quantify the prospective impact of any particular $\theta$ on our posterior estimate, we look to Bayesian active learning, and acquisition functions such as EIG. EIG considers the reduction in uncertainty of model parameters under the inclusion of new data in

expectation over the predictive posterior. Adapting Eq. (1) to our NPE context gives

$$\operatorname{argmax}_\theta \mathbb{H}(\phi|D) - \mathbb{E}_{x \sim p(x|\theta, D)}\Big[\mathbb{H}[\phi|D \cup \{(\theta, x)\}]\Big],$$

which seeks to drive down uncertainty in NDE parameters $\phi$ by optimizing for $\theta$ with simulation outputs $x \sim p(x|\theta, D)$ expected to be most informative. Unfortunately, EIG and related approximations require $p(x|\theta, D)$, which we cannot evaluate nor do we directly approximate in the NPE setting. This makes it considerably more difficult to quantify the utility of candidate $\theta$ values, as we have no direct means of sampling likely simulation outputs.

### 3.2 Characterizing posterior uncertainty

Instead of relying on the marginal distribution $p(x|\theta, D)$, we seek instead to capture uncertainty across different parameterizations of the NDE. Broadly speaking, we want to simulate $\theta$ expected to be informative to our NDE model, reducing epistemic uncertainty as measured by $p(\phi|D)$ and further elucidating parameter sets $\phi$ likely to explain the probabilistic mapping from simulation outputs $x$ to parameter inputs $\theta$. Given that $q_\phi$ models this relationship as the conditional distribution $q_\phi(\theta|x)$, we consider the uncertainty over *distributional* estimates induced by $p(\phi|D)$. This allows for the targeting of epistemic uncertainty in the NDE model according to how that uncertainty appears across feasible target posterior forms $q_\phi(\theta|x)$. More precisely, for some divergence measure $\mathcal{D}(\cdot||\cdot)$, we represent this distributional uncertainty as

$$\mathcal{H}_\mathcal{D}(x) = \mathbb{E}_{\phi|D}[\mathcal{D}\left(p(\theta|x, D)||p(\theta|x, \phi)\right)], \tag{2}$$

where $p(\theta|x, \phi)$ is the corrected posterior produced by $q_\phi(\theta|x)$ (see Section 2.1), and $p(\theta|x, D)$ is the NDE's marginal posterior (under model parameters $\phi$) for simulation posterior estimates (parameters $\theta$):

$$p(\theta|x, D) = \int_\Phi p(\theta|x, \phi)p(\phi|D)d\phi.$$

Intuitively, $\mathcal{H}_\mathcal{D}$ captures a notion of dissimilarity between the posterior estimates from different draws of $\phi \sim p(\phi|D)$. Put another way, $\mathcal{H}_\mathcal{D}$ indicates how certain the NDE is in assigning likelihood values across $\theta \in \Theta$ under a chosen $x$; a relatively low value would indicate that likely parameter draws $\phi \sim p(\phi|D)$ produce posterior estimates $p(\theta|x, \phi)$ that tend to agree with the "marginal" posterior $p(\theta|x, D)$, for instance. Note that when we let the divergence measure be the KL divergence $\mathcal{D} = \mathcal{D}_{\mathrm{KL}}$, we have $\mathcal{H}_{\mathcal{D}_{\mathrm{KL}}} = \mathbb{I}[\phi; \theta|x_o, D]$ (see proof A.1). Computing this exactly is difficult in practice, however, and we therefore seek a practically appropriate approximation below.

### 3.3 Acquisition of informative parameters

Eq. (2) provides a basis for evaluating uncertainty in an NDE without relying on access to or approximations of the likelihood $p(x|\theta)$. Given the posterior estimation problem at hand, we're particularly interested in how to select simulation parameters $\theta$ expected to reduce $\mathcal{H}_\mathcal{D}$ at or around $x_o$. Here we take inspiration from [23], who seek to drive down uncertainty at $\theta$ with noisy estimates of the log joint probability (albeit in a context where likelihoods $p(x|\theta)$ are available). While $\mathcal{H}_\mathcal{D}$ provides a measure of distributional uncertainty, we can target specific $\theta$ whose assigned likelihood is widely disagreed upon across draws of $\phi \sim p(\phi|D)$:

$$\theta^* = \operatorname{argmax}_\theta \mathbb{E}_{\phi|D}\left[\left(p(\theta|x_o, D) - p(\theta|x_o, \phi)\right)^2\right]. \tag{3}$$

While optimization of Eq. (3) over all $\theta$ in the prior support may be ideal, this is computationally infeasible given the posterior estimates from all parameters $\phi \sim \phi|D$ required in expectation. Additionally, note that Eq. (3) does not explicitly account for relative likelihoods of $\theta$ under the posterior or available approximations, possibly leading to $\theta^*$ with high uncertainty under $x_o$ but with low-likelihood under $p(\theta|x_o)$ in expectation. We account for this explicitly during integration with specific SNPE approaches, as seen in the section below.

**Algorithm 1** Active Sequential Neural Posterior Estimation (ASNPE)

---

**Input:** Prior $p(\theta)$, target observation $x_o$, round-wise selection size $B$, round-wise sample size $N$, total rounds $R$
**Output:** Approximate posterior $q_\phi(\theta|x_o)$

   Let $D^{(0)} = \{\}$
   Let $\tilde{p}(\theta) = p(\theta)$
   **for** $r \in [1, \ldots, R]$ **do**
      Draw $N$ samples $\{\theta_i\}_{1:N} \sim \tilde{p}(\theta)$
      Sort $\{\theta_i\}_{1:N}$ by expected divergence (Eq (4)), select top-$B$ $\{\theta_i\}_{1:B}$
      **for** $b \in [1, \ldots, B]$ **do**
         Simulate $x_b \sim p(x|\theta_b)$
         Set $D^{(r-1)} = D^{(r-1)} \cup \{(\theta_b, x_b)\}$
      **end for**
      Set $D^{(r)} = D^{(r-1)}$
      Train NDE $q_\phi$ on $D^{(r)}$: $\phi^* = \text{argmin}_\phi - \sum_{(\theta_i, x_i) \in D^{(r)}} \log \tilde{q}_\phi(\theta_i|x_i)$
      Let $\tilde{p}(\theta) = q_\phi(\theta|x_o)$
   **end for**

---

## 3.4 Integration with APT

In order to tractably approximate Eq. (3), we impose two additional restrictions to bring the parameter acquisition into the SNPE loop:

1. Require the NDE be updated according to APT [17], i.e., trained via maximum likelihood on $\tilde{\mathcal{L}}(\phi) = - \sum_{i=1}^{N} \log \tilde{q}_\phi(\theta_i|x)$, where

$$\tilde{q}_\phi(\theta|x) = q_\phi(\theta|x)\frac{\tilde{p}(\theta)}{p(\theta)}\frac{1}{Z(x,\phi)},$$

   and by *Proposition 1* of [33] ensures $q_\phi(\theta|x) \to p(\theta|x)$ as $N \to \infty$ without requiring post-hoc updates to the NDE's distributional estimate. This allows the model parameter posterior $p(\phi|D)$ to be used directly in the contexts of Eq. (2) and Eq. (3), whereas otherwise the corrective terms involved would need to be accounted for explicitly.

2. To account for the likelihood of $\theta$ under the posterior estimate as captured by the proposal prior $\tilde{p}(\theta) \approx p(\theta|x = x_o)$ in a given round of SNPE, we adjust Eq. (3):

$$\alpha(\theta, p(\phi|D)) = \tilde{p}(\theta) \cdot \mathbb{E}_{\phi \sim \phi|D}\left[(p(\theta|x_o, D) - p(\theta|x_o, \phi))^2\right]. \tag{4}$$

   Further, in practice we approximate this by optimizing Eq. (4) over samples $\theta \sim \tilde{p}^{(r)}(\theta)$, straightforwardly integrating the acquisition mechanism into the standard SNPE pipeline. Round-wise proposals $\tilde{p}^{(r)}(\theta) = q_\phi(\theta|x)$ are set after $N = rB$ samples are collected (for round $r$ with $B$ samples collected per round), and $q_\phi(\theta|x) \to p(\theta|x)$ as $N \to \infty$. All round-wise proposal distributions share the support of the prior $p(\theta)$, which itself is established as having support over the entire parameter domain of interest $\Theta$.

   Thus, optimizing $\alpha$ over a sample of size $N$ drawn from a proposal distribution $\tilde{p}^{(r)}(\theta)$ at any round $r$ recovers the true optimum of $\alpha$ as $N \to \infty$; each proposal's support connects back to the prior's support, which covers $\Theta$. As a result, at each round a fixed sample size $N$ drawn from the proposal can be used to approximate the acquisition maxima, while additionally adhering to the round-wise proposal sampling required to ensure $q_\phi(\theta|x) \to p(\theta|x)$. Refer to Section A.2 for additional discussion on the functional form of the acquisition function.

3. (Optional, depending on model) To approximate the Bayesian model parameter posterior $p(\phi|D)$, neural network-based NDEs (such as flow-based generative models or mixture density networks) can be trained via MC-dropout [21, 14]. See additional details regarding consistent sampling and log probability evaluation in MAFs under MC dropout in Appendix C.

Altogether, this constitutes the ASNPE method, which is more succinctly described in Algorithm 3.3. See Figure 1 for a visual depiction of this process.

### 3.5  Bayesian origin-destination calibration

We now position OD calibration as a Bayesian inference problem, with a posterior density of interest to be approximated by SNPE methods. During a time interval of interest $[t_s, t_e]$ on a traffic network $G$, we consider a single OD matrix $d = \{d_z\}_{z \in \mathcal{Z}}$, where $d_z$ represents the expected travel demand for the origin-destination pair $z$. $\mathcal{Z}$ is the set of OD pair indices, i.e., $\mathcal{Z} = \{1, 2, \ldots, Z\}$, for all pairs of interest on $G$. OD pairs are typically defined between elements in a fixed set of Traffic Assignment Zones (TAZs) whose size may not be uniform due to variable demand density; see Figure 7 for zones drawn on two candidate networks. Figure 2 loosely depicts the acquisition pipeline for traffic data, corresponding the collection process shown in Figure 1.

Conventionally, OD calibration is formulated as a simulation-based optimization problem over a traffic simulator $\mathcal{S}(\cdot; u_1, u_2)$, where $u_1, u_2$ are vectors of endogenous simulation variables and exogenous simulation parameters, respectively. The goal is to obtain an OD matrix $d^*$ that yields simulation results $x^* = \mathcal{S}(d^*; u_1, u_2)$ that are sufficiently close to available observational data $x_o$.

While many pre-existing methods adopt a traditional optimization scheme and iteratively produce point estimates for $d^*$, we formulate the calibration problem under the Bayesian paradigm and instead seek a posterior $p(d|x; \hat{d})$

$$p(d|x; \hat{d}) = \frac{p(x|d)p(d; \hat{d})}{p(x; \hat{d})} = \frac{p(x|d)p(d; \hat{d})}{\int p(x|d)p(d; \hat{d})dd} \quad (5)$$

where $p(d; \hat{d})$ represents the prior distribution over OD matrices, often defined around a noisy historical estimate $\hat{d}$. The posterior estimate under our observation $p(d|x = x_o; \hat{d})$ can then be used to compute different point estimates for $d^*$, used in other downstream tasks as an informative prior, and can represent intrinsic uncertainty in the calibration problem. This formulation achieves parity with existing approaches, where $\hat{d}$ is otherwise used as a noisy starting point. Additionally, the traffic simulator $\mathcal{S}$ is treated as a black-box that implicitly defines the likelihood $p(x|d)$:

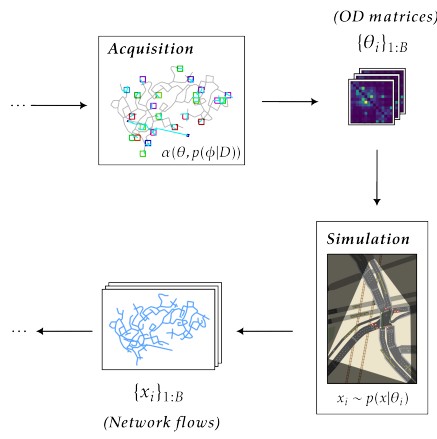

Figure 2: Simple depiction of the data acquisition and simulation process for the OD calibration application. The acquisition step selects parameter candidates (OD matrices) to then be simulated (via SUMO) and produce outputs (network flow observations) that are used to update the approximate posterior model.

$$p(x|d) = \int p_{\mathcal{S}}(x, z|d)dz = \int p_{\mathcal{S}}(x, u_1, u_2|d)du_1 du_2, \quad (6)$$

i.e., marginalizing over all possible latent trajectories $z$. As is typical in simulation-based inference settings, this integral is intractable for simulators of sufficient complexity.

## 4  Experimental results

We explore the performance of the proposed ASNPE method in the context of OD calibration on a challenging real-world traffic network. Our goal here is to 1) compare the general purpose utility of our approach in complex settings against tuned benchmark methods, and 2) verify ASNPE's candidacy as a sample efficient posterior estimation tool in high-dimensional, data-scarce settings. These objectives are directly in line with the needs of practitioners, both in the urban mobility community and broadly across scientific disciplines.

### 4.1  Experimental setup

We conducted a case study on the large-scale regional Munich network seen in [37]. The Munich network includes 5329 configurable origin-destination pairs (constituting simulation input), as

| | | Hours 5:00-6:00 | | Hours 8:00-9:00 | |
| --- | --- | --- | --- | --- | --- |
| | | Cong. level A | Cong. level B | Cong. level A | Cong. level B |
| | Prior OD | 0.178 | 0.165 | 0.181 | 0.150 |
| **Prior I** $r = 0.6$ $q = 0.3$ | Setting prior | 0.396 | 0.488 | 0.539 | 0.387 |
| | SPSA | $0.563 \pm 0.089$ | $0.521 \pm 0.052$ | $0.453 \pm 0.078$ | $0.384 \pm 0.049$ |
| | PC-SPSA | $0.193 \pm 0.063$ | $0.185 \pm 0.097$ | $\mathbf{0.159} \pm 0.036$ | $\mathbf{0.159} \pm 0.046$ |
| | MC-ABC | $0.275 \pm 0.047$ | $0.295 \pm 0.066$ | $0.343 \pm 0.036$ | $0.305 \pm 0.036$ |
| | SNPE | $0.201 \pm 0.085$ | $0.167 \pm 0.092$ | $0.187 \pm 0.059$ | $0.314 \pm 0.025$ |
| | (ours) ASNPE | $\mathbf{0.147} \pm 0.011$ | $\mathbf{0.157} \pm 0.097$ | $0.165 \pm 0.064$ | $0.161 \pm 0.079$ |
| **Prior II** $r = 0.75$ $q = 0.45$ | Setting prior | 0.340 | 0.311 | 0.245 | 0.277 |
| | SPSA | $0.316 \pm 0.074$ | $0.342 \pm 0.045$ | $0.258 \pm 0.061$ | $0.189 \pm 0.025$ |
| | PC-SPSA | $0.180 \pm 0.029$ | $0.189 \pm 0.055$ | $0.163 \pm 0.032$ | $0.155 \pm 0.031$ |
| | MC-ABC | $0.143 \pm 0.034$ | $0.190 \pm 0.036$ | $0.169 \pm 0.023$ | $0.140 \pm 0.010$ |
| | SNPE | $0.137 \pm 0.025$ | $0.157 \pm 0.032$ | $0.142 \pm 0.024$ | $0.135 \pm 0.016$ |
| | (ours) ASNPE | $\mathbf{0.130} \pm 0.024$ | $\mathbf{0.148} \pm 0.034$ | $\mathbf{0.138} \pm 0.025$ | $\mathbf{0.132} \pm 0.016$ |

Table 1: RMSNE scores on the Munich traffic network, as described in Section 4.1. Note that methods like SPSA (poor convergence aside) can produce RMSNE scores larger than the reported setting prior due to noise in the starting sample. The "setting prior" value is an average RMSNE score over many $\theta$ draws from the shifted prior. Reported errors are empirical standard deviations computed over the five trial runs.

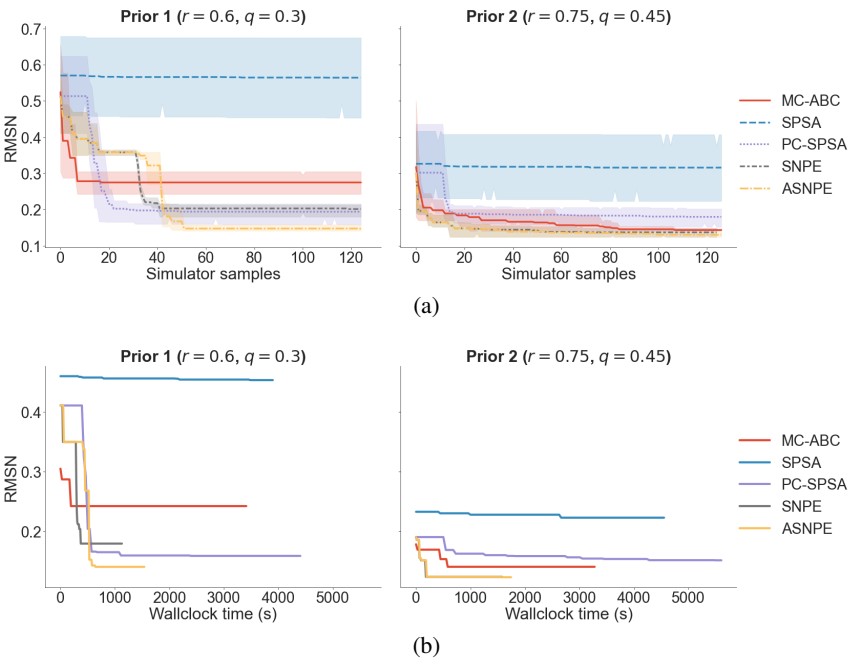

(a)

(b)

Figure 3: Plots of the (averaged) calibration horizons for each of the evaluated methods on the *Prior I, Hours 5:00-6:00, Congestion level A* scenario. **(a)** RMSN(E) scores reached throughout the 128 sample simulation horizon for each evaluated method, averaged over five repeated trials (mean line plotted) and with error bars calculated as bootstrapped 95% confidence intervals. **(b)** The same scores shown in *(a)*, but instead plotted against the wallclock time passed before the score was reached (for each method's single best run). Note that the full 128-sample method trajectories are included, and the variability in line lengths demonstrates both 1) the impact of NPE-based methods' ability to run simulations in parallel, and 2) noisiness in simulation runtimes due to the variable inputs explored by each method. See Appendix E for all scenario plots.

|  | Method | C2ST | MMD | MED-DIST | MEAN-ERR |
|---|---|---|---|---|---|
| *Bernoulli GLM* | SNPE-C | $0.749 \pm 0.017$ | $0.210 \pm 0.024$ | $\mathbf{11.454} \pm 0.255$ | $0.188 \pm 0.117$ |
|  | ASNPE | $\mathbf{0.725} \pm 0.012$ | $\mathbf{0.146} \pm 0.057$ | $11.993 \pm 0.172$ | $\mathbf{0.150} \pm 0.085$ |
| *SLCP distractors* | SNPE-C | $0.987 \pm 0.001$ | $0.172 \pm 0.001$ | $16.716 \pm 1.014$ | $\mathbf{0.899} \pm 0.065$ |
|  | ASNPE | $\mathbf{0.985} \pm 0.002$ | $\mathbf{0.148} \pm 0.022$ | $\mathbf{16.547} \pm 0.499$ | $0.906 \pm 0.220$ |
| *Gaussian mixture* | SNPE-C | $0.773 \pm 0.009$ | $0.167 \pm 0.006$ | $1.051 \pm 0.037$ | $0.532 \pm 0.074$ |
|  | ASNPE | $\mathbf{0.771} \pm 0.006$ | $\mathbf{0.150} \pm 0.025$ | $\mathbf{1.010} \pm 0.066$ | $\mathbf{0.440} \pm 0.164$ |

Table 2: Results comparing SNPE-C and ASNPE for various metrics on the *Bernoulli GLM*, *SLCP distractors*, and *Gaussian mixture* tasks from [28]. Experimentation details, metrics, and associated plots can be found in Appendix B.

well as 507 detector locations (positions of reported output traffic flows), resulting in a highly underdetermined system.

We build and evaluate a number of synthetic demand scenarios, following an established framework for fair evaluation of urban demand calibration methods ([2, 39, 9]). Each of the test demand scenarios are constructed from combinations of the following factors:

**Time interval**: a time interval of interest is specified through which to simulate traffic flows on the network. Prior ODs are chosen to reflect real-world traffic patterns for the affiliated times. We evaluate peak morning demand for hour-long intervals at 5:00am-6:00am and 8:00am-9:00am.

**Congestion level**: within a given time interval, we can further control the level of traffic congestion exposed during the hour. The distribution of frequencies present in the starting ODs plays a critical role in determining route time across the traffic network. Here we define two congestion levels, "A" and "B," to reflect different average frequencies assigned to OD pairs. Here we use a truncated normal distribution (lower bound at 0) to sample OD counts with varying means and variance: **(1)** ($\mu = 5, \sigma = 25$) for *hours 5:00-6:00, congestion level A*, **(2)** ($\mu = 10, \sigma = 50$) for *hours 5:00-6:00, congestion level B*, **(3)** ($\mu = 25, \sigma = 50$) for *hours 8:00-9:00, congestion level A*, **(4)** ($\mu = 50, \sigma = 100$) for *hours 8:00-9:00, congestion level B*.

**Prior bias and noise**: under each time interval and congestion level, we further perturb the generated OD matrices to represent realistic variance found in real-world sampling of traffic observations. Here we use the following noise model, mirroring that of [39]: $x_c = (r + q \times \delta) \times \hat{d}$, where $\delta \sim N(0, \frac{1}{3})$. We then formulate two perturbed settings: 1) *Prior I: r = 0.6, q = 0.3*, and 2) *Prior II: r = 0.75, q = 0.45*. Prior I constitutes a heavily under-congested estimate with relatively little added noise, while Prior II is less biased from the true OD but noisier. Both priors represent underestimations of the true demand, reflecting the fact that most prior ODs from real-world settings are constructed from historic travel demand observations.

The eight synthetic combinations constitute starting ODs/priors that span a variety of different settings important for real-world urban demand calibration tasks. Each synthetic setting yields a particular prior OD estimate $\hat{d}$, which is then used to construct a prior $p(d; \hat{d})$. A fixed sample is drawn from $p(d; \hat{d})$ and passed through the open-source traffic simulator Simulation of Urban MObility (SUMO) [26] to generate an associated "true" network flow $x_o$.

## 4.2 Comparison to SOTA Calibration Methods

We evaluate the effectiveness of the proposed solution by comparing against available SOTA benchmarks commonly employed in the OD calibration space: Simultaneous Perturbation Stochastic Approximation (SPSA) [42] and principal component (PC)-based SPSA, or PC-SPSA [37]. SPSA is a widely employed algorithm for travel demand calibration, and PC-SPSA is an effective extension that optimizes over parameters in a lower-dimensional subspace, as defined by the principal components of computed travel demand history matrix. Both of these methods are conventional optimization-based methods, and do not leverage neural networks. Additionally, these methods in their canonical form cannot be parallelized, requiring serial simulation evaluations across each iteration.

For NPE-based approaches, we evaluate our proposed method ASNPE alongside SNPE-C (or APT)[17] and Approximate Bayesian Computation (ABC)[41, 6]. ABC serves primarily as a less

sophisticated baseline that reflects early approaches to likelihood-free inference, and it typically faces difficulty scaling and is far less flexible compared to its NPE counterparts.

For all of the eight scenario priors $p(d; \hat{d})$, each calibration method is ran for a maximum of 128 SUMO simulations, seeking to recover $x_o$. The root mean squared normalized error (RMSNE) is recorded for each method's simulation horizon, as used in [39] (see also Appendix A.3). To account for the stochasticity across evaluations, we report RMSNE averaged of five repeated simulation runs. See Table 1 for reported values for each method across each of the eight synthetic scenarios, as well as paired prior plots in Figure 3.

### 4.3 Analyzing calibration performance

**ASNPE outperforms all other methods across most explored settings**: as can be seen in Table 1, our method outperforms both the well-tuned PC-SPSA method commonly employed by the urban mobility community, as well as popular simulation-based inference (SBI) methods like SNPE, across almost all of the explored settings. In general, PC-SPSA tends to quickly converge but demonstrates a limited ability to further improve beyond the first 10-20 encountered simulations. Both SBI methods tend to make steady improvements throughout the entire trial, however, albeit often doing so more slowly in the first 20-40 simulations than PC-SPSA. This is primarily due to the limited feedback SNPE/ASNPE receive comparatively, only incorporating new simulation data in batches (in this case, every 32 simulation draws).

Additionally, ASNPE reliably reaches better RMSNE scores than SNPE with fewer simulations, as well as Approximate Bayesian Computation (MC-ABC). This can be seen as early as the first NDE update, before which the two methods encounter the same (seeded) simulation samples. This also empirically supports our central methodological contribution, i.e., optimization over informative simulation parameters can more efficiently improve the accuracy of the inferred posterior estimate.

**ASNPE is outperformed in some cases**: ASNPE is outperformed by PC-SPSA in two of our explored settings (under Prior I, Hours 8:00-9:00). While we wouldn't expect a single method to be the best choice for all variations in such a high-dimensional setting, this particular scenario serves as an opportunity to better understand possible failure modes of the proposed method.

As alluded to above, ASNPE updates its internal model only after a batch of simulation samples is generated, whereas PC-SPSA adjusts its parameters after each simulation run. While generating samples in batches can be beneficial (and is often necessary) for early stability of ASNPE, it can mean informative simulation data is incorporated later in the trial. This explains the occasional gap that opens up between SBI methods and PC-SPSA in the first 30 simulations, only after which is ASNPE/SNPE able to incorporate the samples to improve its posterior estimate. Note, however, that most of the early advantage PC-SPSA may have over ASNPE is dwarfed by the ability to obtain its simulation draws in parallel, whereas PC-SPSA must run simulations serially. This allows for larger, more stable improvements in less time, which can be seen in subplot (b) of Figure 3.

### 4.4 Performance on common SBI benchmarks

We additionally report results on several common SBI benchmark environments, and compare against the performance of (non-active) SNPE. Numerical results can be found in Table 2, along with plots and more details in Appendix B. These additional results demonstrate the wider applicability of our method beyond the travel demand calibration task.

## 5 Conclusion

In this paper, we introduced Active Sequential Neural Posterior Estimation (ASNPE), an SNPE variant that actively incorporates informative simulation parameters $\theta$ to drive down epistemic uncertainty in the neural density estimator and improve sample efficiency for high-quality estimates. We evaluate this method on a complex, high-dimensional problem in urban demand calibration, and show it reliably outperforms available benchmark methods across a variety of scenarios with variable bias and noise. We additionally provide results on several common SBI benchmark environments, and find ASNPE is capable of outperforming state-of-the-art SNPE methods on key posterior approximation metrics.

# 6 Acknowledgements

This work was supported in part by the National Science Foundation under awards #2226087 and #1837131.

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

# A  Appendix

## A.1  Connection between mutual information and $\mathcal{H}_{\mathcal{D}_{\mathrm{KL}}}$

Expanding the definition of mutual information for $\mathbb{I}[\phi; \theta | x_o, D]$, we have the following (note that $x_o$ and $D$ are fixed):

$$
\begin{aligned}
\mathbb{I}[\phi; \theta | x_o, D] &= \int_\Phi \int_\Theta p(\theta, \phi | x_o, D) \log\left(\frac{p(\theta, \phi | x_o, D)}{p(\theta | x_o, D) p(\phi | x_o, D)}\right) d\theta d\phi \\
&= \int_\Phi \int_\Theta p(\theta | \phi, x_o, D) p(\phi | x_o, D) \log\left(\frac{p(\theta | x_o, \phi) p(\phi | x_o, D)}{p(\theta | x_o, D) p(\phi | x_o, D)}\right) d\theta d\phi \\
&= \int_\Phi p(\phi | x_o, D) \left[\int_\Theta p(\theta | x_o, \phi) \log\left(\frac{p(\theta | x_o, \phi)}{p(\theta | x_o, D)}\right) d\theta\right] d\phi \\
&= \int_\Phi p(\phi | x_o, D) \left[\mathcal{D}_{KL}\left(p(\theta | x_o, \phi) || p(\theta | x_o, D)\right)\right] d\phi \\
&= \mathbb{E}_{p(\phi | x_o, D)}\left[\mathcal{D}_{KL}\left(p(\theta | x_o, \phi) || p(\theta | x_o, D)\right)\right]
\end{aligned}
$$

This aligns with the expected divergence term introduced in Eq. 2 when we let the measure $\mathcal{D} = \mathcal{D}_{\mathrm{KL}}$. The connection to mutual information helps to position our motivation for the acquisition function ultimately introduced in Eq. 4. That is, by seeking to drive down disagreement between "marginal" and "component" posteriors $p(\theta | x, D)$ and $p(\theta | x, \phi)$, respectively, we are attempting to realize the information we expect $\theta$ to tell us about our NDE parameters $\phi$, thereby minimizing the remaining mutual information between the two.

## A.2  Regarding the functional form of the acquisition function

The family of functions $\alpha(\theta, p(\phi | D)) = \tilde{p}(\theta)(\mathbb{E}_{\phi | D}[\dots])^\lambda$ under parameter $\lambda$ constitutes valid choices for the acquisition function for any $\lambda$, facilitating different levels of emphasis on uncertainties at values of $\theta$ relative to their likelihoods under $\tilde{p}$. While several values of $\lambda$ may be justifiable, the choice to use $\lambda = 1$, implicit in Eq. 4, intuitively captures a desirable balance in the relationship between uncertainties and likelihoods of $\theta$.

In particular, under level sets $\alpha(\cdot, p(\phi | D)) = z$ (where $p(\phi | D)$ is held constant), as likelihoods $\tilde{p}(\theta)$ decrease by a factor of $n$, the average deviation between $p(\theta | x, D)$ and $p(\theta | x, \phi)$ need only increase by a factor of $\sqrt{n}$, i.e., changes in uncertainty are sub-linear in the likelihood ratio. With the introduction of variable $\lambda$, this factor generalizes to $n^{1/(2\lambda)}$, and may require additional measures to balance the resulting sensitivity between the terms. We find that $\lambda = 1$ is a natural choice that reasonably captures the desire to explore potentially unlikely parameters with high uncertainties without ignoring them (e.g., $\lambda \to 0$) or relying too heavily on them (e.g., $\lambda \to \infty$).

## A.3  Additional Definitions

The root mean squared normalized error (RMSNE) for a simulated output $\hat{x}$ with respect to an observational reference $x_o$, as used in [39], is defined as

$$
\mathrm{RMSNE} = \frac{\sqrt{n \sum_{i=1}^n (\hat{x}^{(i)} - x_o^{(i)})^2}}{\sum_{i=1}^n x_o^{(i)}},
$$

where there $n$ is the number of observed segment flows, and both $x_o$ and $\hat{x}$ are $n$-dimensional vectors.

# B  Additional SBI benchmarks

In order to appeal to the general utility of our proposed method, we provide additional experimental results between ASNPE and SNPE-C [17] on three common SBI benchmark tasks: *SLCP distractors*, *Bernoulli GLM*, and *Gaussian Mixture*. Each of these settings corresponds to a reproducible task

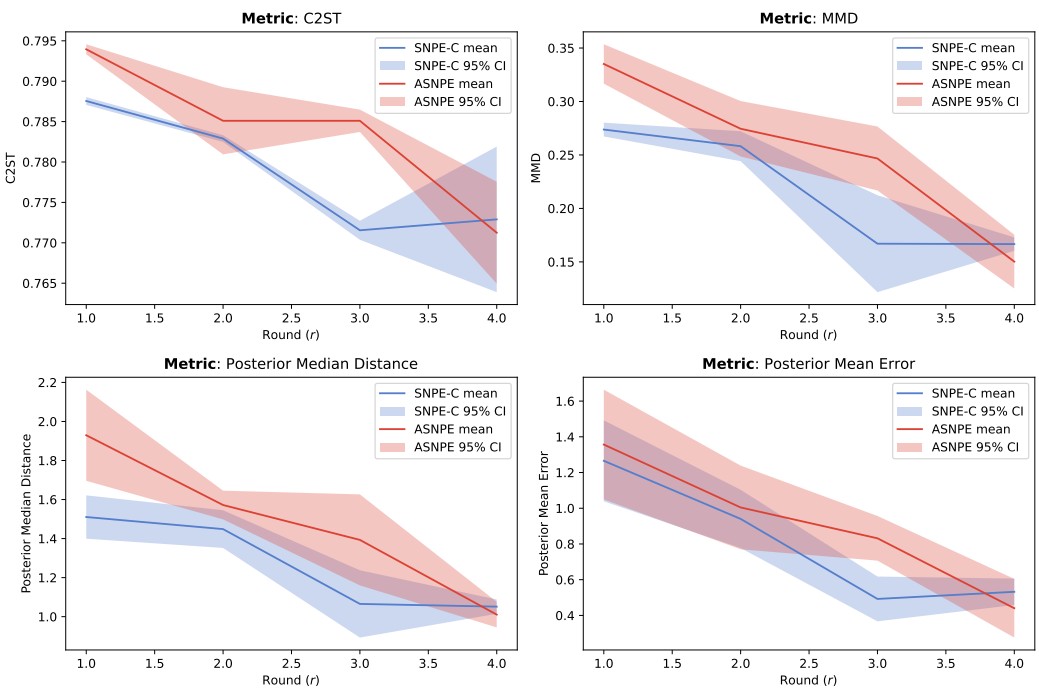

Figure 4: Results on various metrics between ASNPE and SNPE-C across four rounds of sequential inference for the *Gaussian mixture* task.

environment from [28], and the corresponding implementation in the sbibm Python package is used to run our experiments.

We additionally evaluated our method on these tasks using metrics beyond RMSNE, the primary metric used for the travel demand calibration case study. These include classifier 2-sample tests (C2ST), maximum mean discrepancy (MMD), the posterior median distance (median $L^2$ norm between simulated samples $\theta_i \sim p(\theta|x_o) \to x_i \sim p(x|\theta_i)$ and the observation $x_o$), and the posterior mean error (normalized absolute error between the true posterior mean and the approximate posterior mean). Full details for each of these tasks and metrics can be found in [28].

We evaluate both methods over medium-size sample horizons: 4 rounds with 256 samples per round, for a total of 1024 simulation samples. Note that this is eight times larger than the sample sizes collected for the trials on the travel demand task. For reference, 256 samples in our (non-parallelized) SUMO environment takes ~6 hours, whereas 256 samples from the SLCP simulator takes ~10 seconds on our hardware.

Trials were repeated five times for each method, and the average score and standard deviation for each metric over these trials are shown in Figures 4, 5, 6. Note that smaller values are better for each metric (C2ST ranges between $0.5 - 1.0$). While these simulation horizons are relatively small, we find that, by the final round, ASNPE tends to outperform SNPE across most settings and on most metrics. In particular, ASNPE outperforms SNPE on C2ST and MMD across all settings, along with the distance-based metrics on all but the median distance for Bernoulli GLM and mean error for SLCP Distractors.

While SNPE-C is a state-of-the-art benchmark method, comparing against it also constitutes an ablation test for ASNPE's acquisition component. Although parameter sets are chosen differently and the underlying NDE varies (minimally to accommodate the need to approximate $p(\phi|D)$) across methods, the sequential inference procedures are otherwise identical. These results help to isolate and identify the contribution of the active learning scheme across a wider range of tasks and metrics for the overarching goal of producing accurate posterior approximations holistically (i.e., not just well-calibrated point estimates).

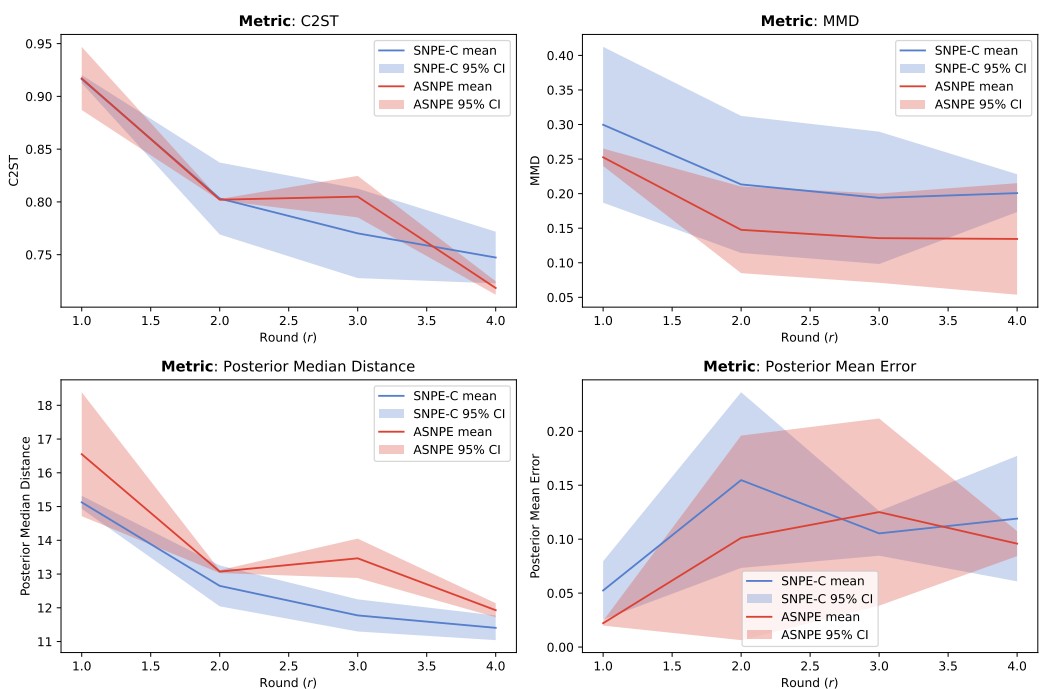

Figure 5: Results on various metrics between ASNPE and SNPE-C across four rounds of sequential inference for the *Bernoulli GLM* task.

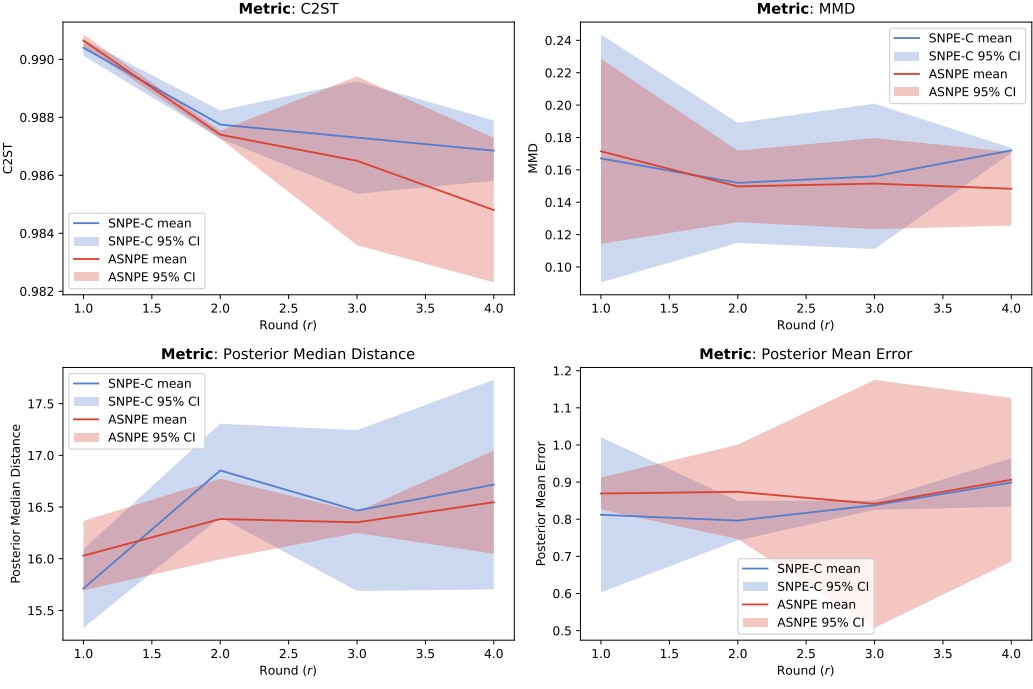

Figure 6: Results on various metrics between ASNPE and SNPE-C across four rounds of sequential inference for the *SLCP Distractors* task.

## C  Code reproducibility

In the spirit of reproducibility and in the hopes our code may be of use for follow-up works, we provide the following Python packages:

1. `seqinf` package: this package includes a full implementation of ASNPE and convenient abstractions around the popular `sbi`[44] and `sbibm`[28] Python packages for general sequential inference pipelines. This package also includes an implementation of the masked autoregressive flow (MAF) with consistent MC-dropout that was used as an NDE for all experiments. Available at `https://github.com/samgriesemer/seqinf`.

2. `sumo_cal` package: short for SUMO calibration, this package includes many general programmatic utilities for calibrating traffic models using results from the SUMO traffic simulator[26] with a high-level Pythonic interface, including collecting results from several runs, running in multi-threaded contexts, employing standardized configuration, etc. See Figure 11 for a snapshot of the SUMO interaction scheme. Available at `https://github.com/samgriesemer/sumo_cal`.

## D  Additional demand calibration experimentation details

### D.1  Descriptions of SOTA Calibration Methods

- SPSA: SPSA (Simultaneous Perturbation Stochastic approximation) is an optimization algorithm for systems with multiple unknown parameters, which can be used for large-scale models and various applications. It can find global minima, like simulated annealing [46]. SPSA works by approximating the gradient using only two measurements of the objective function with gradients, making it scalable for high-dimensional problems [42].

- PC-SPSA: PC-SPSA is proposed to address fundamental scalability issues with SPSA. This is because SPSA searches for the optimal solution in a high-dimensional space without considering the structural relationships among the variables. PC–SPSA combines SPSA with principal components analysis (PCA) to reduce the problem dimensionality and limit the search noise. PCA captures the structural patterns from historical estimates and projects them onto a lower-dimensional space, where SPSA can perform more efficiently and effectively [39].

Specifically, we implement the SPSA and PC-SPSA algorithms according to [38] and associated open-source implementations[4]. In addition, we employ the so-called `Method-6`, titled "Spatial, Temporal, and Day-to-Day Correlation," as detailed in [38]. This provides a means of systematically generating needed historical data for PCA, and according to the original work constitutes the most robust and optimal solution among the proposed variants.

#### D.1.1  Hyperparameter details & computing resources

Here we include a brief discussion on the implications of the hyperparameters found in ASNPE, as well as the settings used in our experiments. Keeping the total number of simulation samples constant,

1. The number of rounds $R$ dictates how many times we update the proposal distribution over the course of the simulation horizon. Increasing this value can enable quicker feedback to the NDE, requiring fewer simulation samples before re-training the model. When the prior is well-calibrated and simulation samples are representative of the observational data, this can have a positive compounding effect that boosts the rate of convergence to the desired posterior. However, for larger $R$ the resulting batch sizes are smaller and the NDE receives noisier updates, which can have the opposite effect and hurt early performance when the prior is poor.

2. The number of selected samples $B$ per round is directly determined by $R$ when the total number of simulations is held constant, and thus the above effects apply here.

---

[4]`https://github.com/LastStriker11/calibration-modeling`

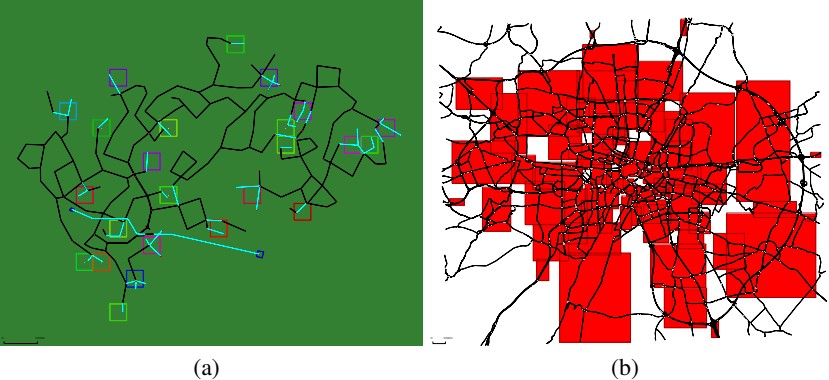

(a)          (b)

Figure 7: Depiction of traffic networks in the SUMO simulator. **(a)** depicts a relatively small synthetic network for reference, with approximately 500 configurable OD pairs. **(b)** shows the larger Munich network used in our reported experimental settings, with an order of magnitude more configurable OD pairs than the synthetic network.

3. The number of proposal samples $N$ per round governs the size of the parameter candidate pool over which the acquisition function is evaluated. Increasing this value allows us to consider more potentially relevant candidates under $\tilde{p}(\theta)$, and can thus increase the quality of the resulting $B$-sized batch. Given the acquisition function can be evaluated over this pool very efficiently (i.e., as a batched inference step through the NDE model), one can practically scale this up arbitrarily to increase the sample coverage over the proposal support (but with decreasing marginal utility).

The following are additional hyperparameter details for the evaluated methods:

1. **In the SNPE loop**: total number of rounds $R$ (4 in reported experiments), round-wise sample size $N$ (between 256-512), round-wise selection size $B$ (32 in reported experiments).

2. **Neural Density Estimator (NDE) model**: our model architecture (used for both SNPE and ASNPE) is a masked autoregressive flow with 5 transform layers, each with masked feedforward blocks containing 50 hidden units, and trained with a (consistent) MC-dropout setting of 0.25. When collecting distributional estimates as described in Eq 4, we used 100 weight samples $\phi \sim p(\phi|D)$ (as generally recommended in [21]).

3. **PC-SPSA**: this method uses PCA to optimize OD estimates in a lower-dimensional subspace of the 5329-dimensional parameter space. The number of the principal components is chosen such that 95% of the variance is recovered in the provided historical OD estimate (which is further dictated by the choice of prior distribution). The number of PCs used by this method across the many explored settings presented in Section 4.1 varies from 99-117.

All experimentation code is written in Python 3.11. To run experiments, we employed our own hardward locally, which is an linux-based machine running an Intel(R) Core(TM) i9-10900X CPU @ 3.70GHz 64GB memory, and NVIDIA GeForce RTX 2080 Ti.

# E    Additional demand calibration plots

Figures 8, 9, and 10 are plots of calibration horizons for the remaining settings of the demand calibration task not shown in the main paper (which highlighted the first scenario, *Prior I, Hours 5:00-6:00, Congestion level A*). The associated RMSNE scores can all be found in Table 1 in the main paper body.

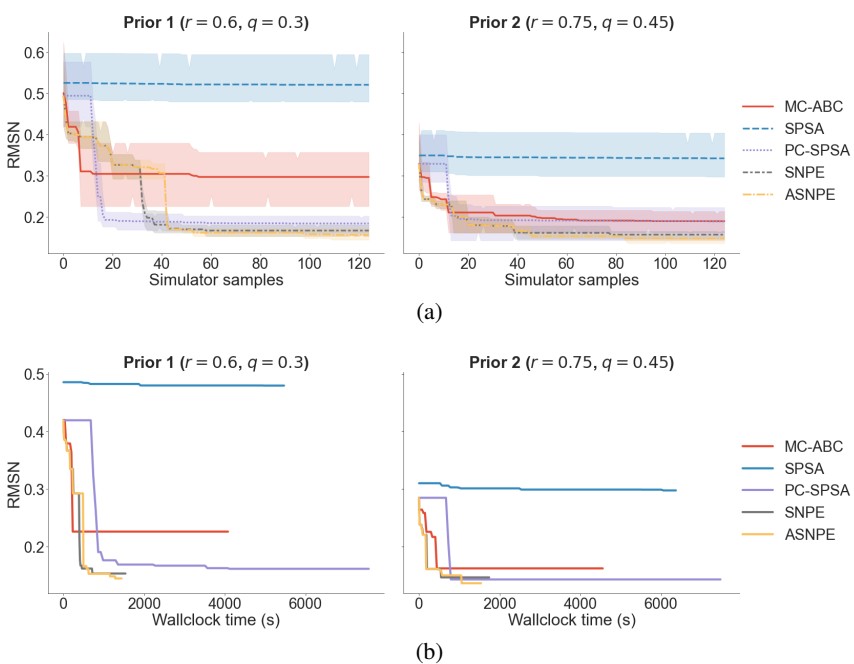

(a)

(b)

Figure 8: Plots of the (averaged) calibration horizons for each of the evaluated methods on the *Prior II, Hours 5:00-6:00, Congestion level B* scenario.

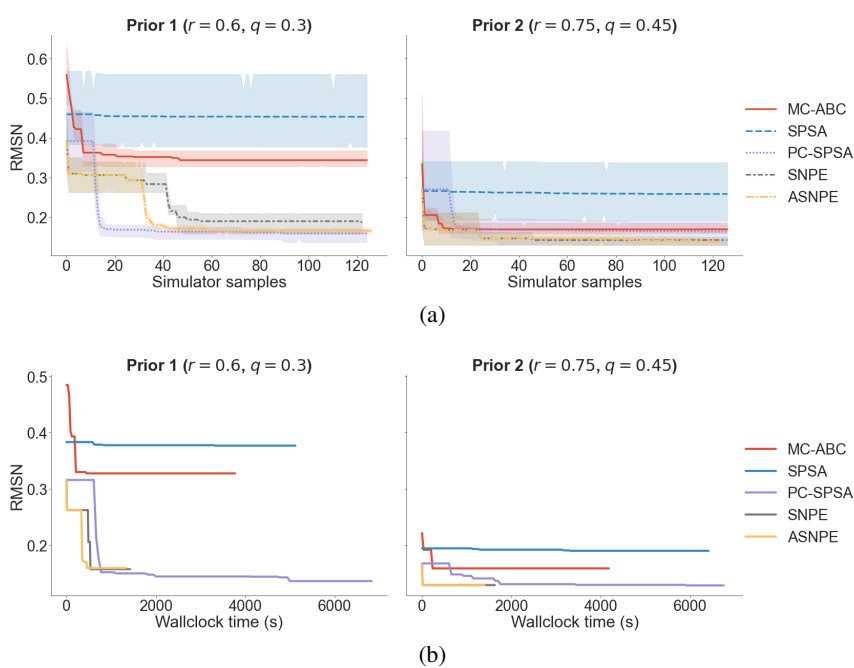

(a)

(b)

Figure 9: Plots of the (averaged) calibration horizons for each of the evaluated methods on the *Prior I, Hours 8:00-9:00, Congestion level A* scenario.

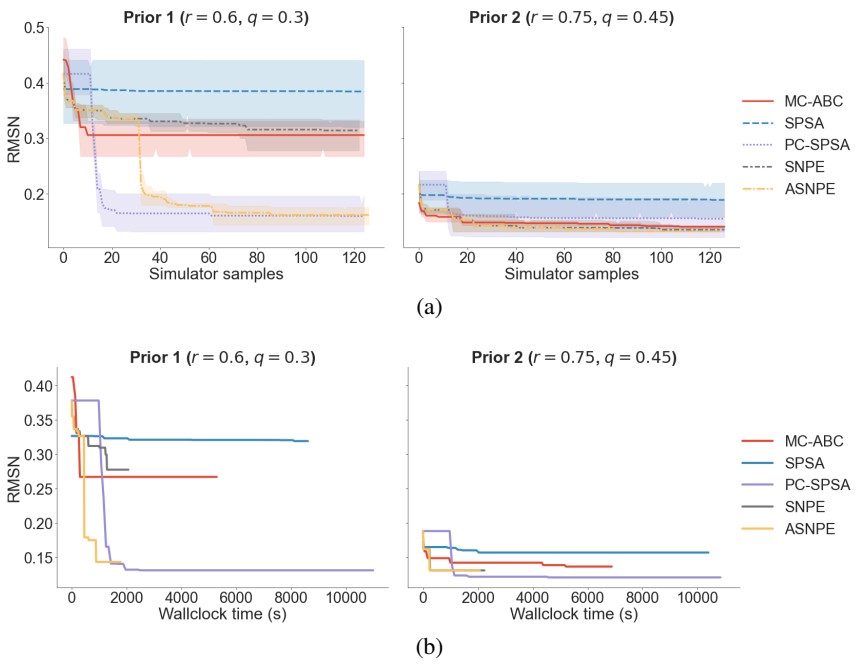

(a)

(b)

Figure 10: Plots of the (averaged) calibration horizons for each of the evaluated methods on the *Prior II, Hours 8:00-9:00, Congestion level B* scenario.

## F Additional distribution plots and schematics for demand calibration

### F.1 Simulation schematic

Figure 11 provides a more detailed look at the programmatic interaction with the SUMO simulator.

### F.2 Additional posterior plots

Figure 12 and figure 13 provide additional plots of the posterior approximation for the primary travel calibration task.

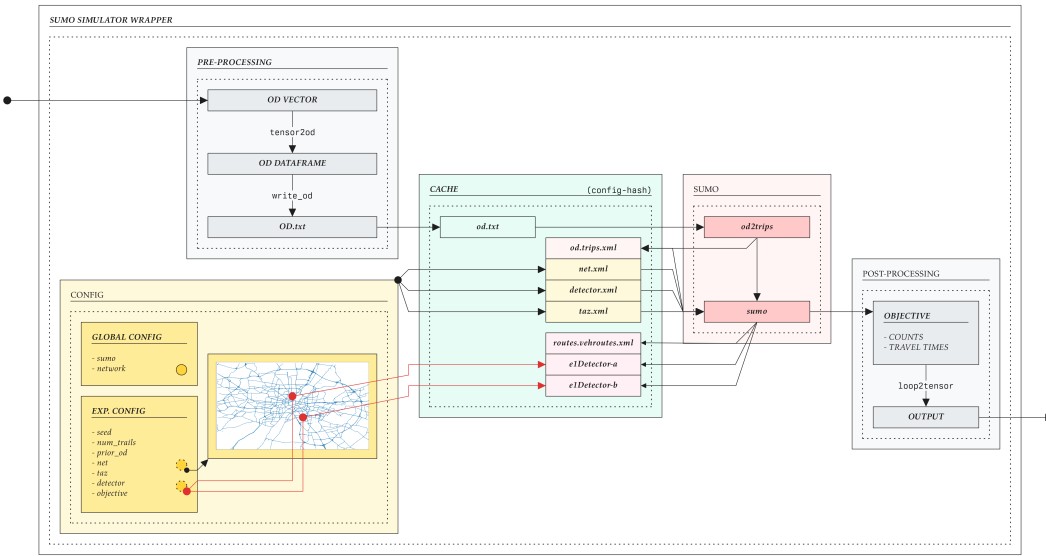

Figure 11: This diagram provides a more detailed look at some of the internal details behind the preparation of input to and the transformation of output from the simulator. An input OD vector $\theta$, drawn from some proposal distribution in the outer method context, 1) "enters" the diagram at the left, 2) is transformed into a suitable representation for SUMO, 3) combined with additional configuration and network files, and 4) run through the SUMO simulator, after which the output is parsed to produce the resulting segment flows $x$ under demand $\theta$.

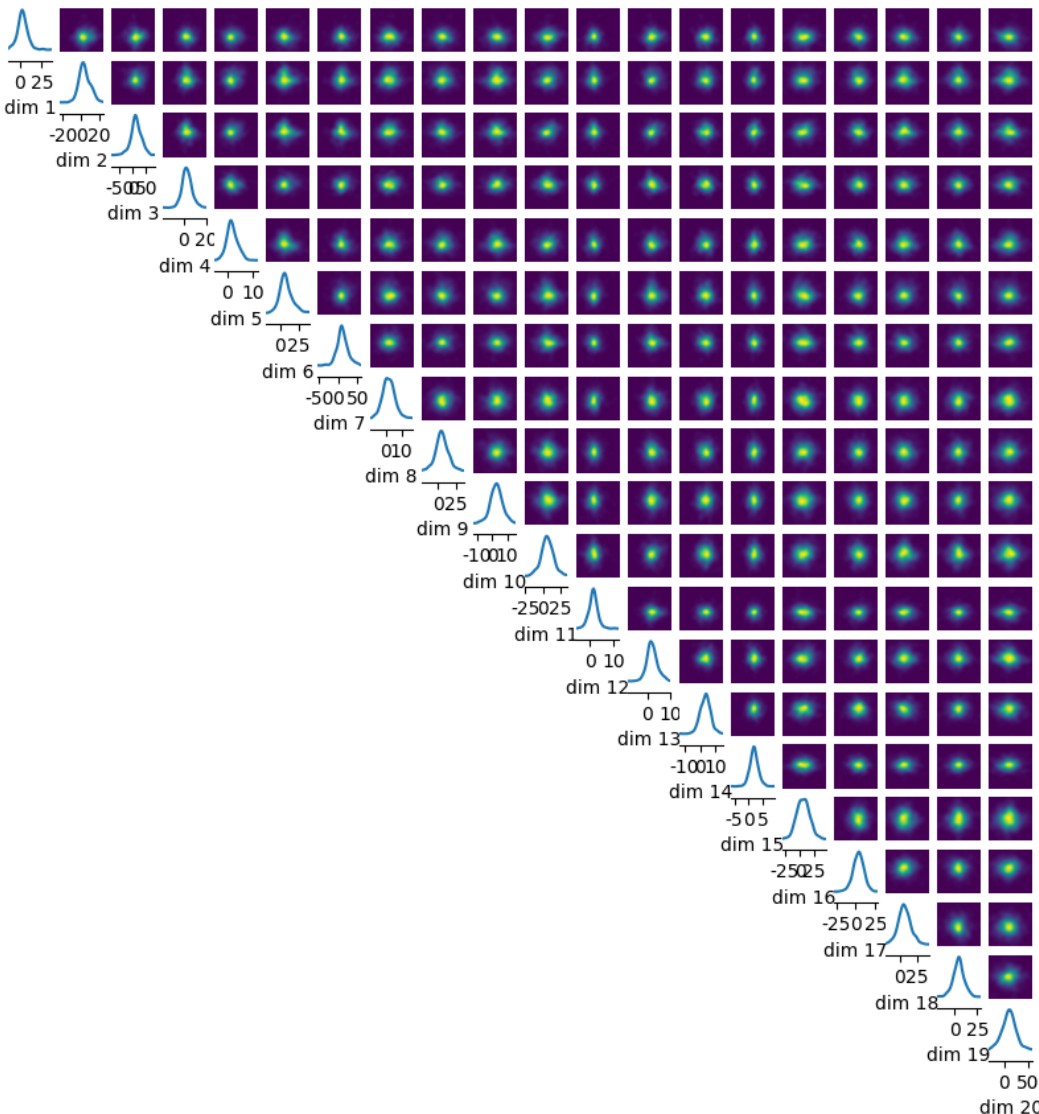

Figure 12: Pairwise density plots of a 20-dimensional slice of the final approximate posterior $\tilde{p}^{(R)}(\theta) = q_\phi(\theta|x_o)$ produced by ASNPE on the *Prior I, Hours 5:00-6:00, Congestion level A* scenario.

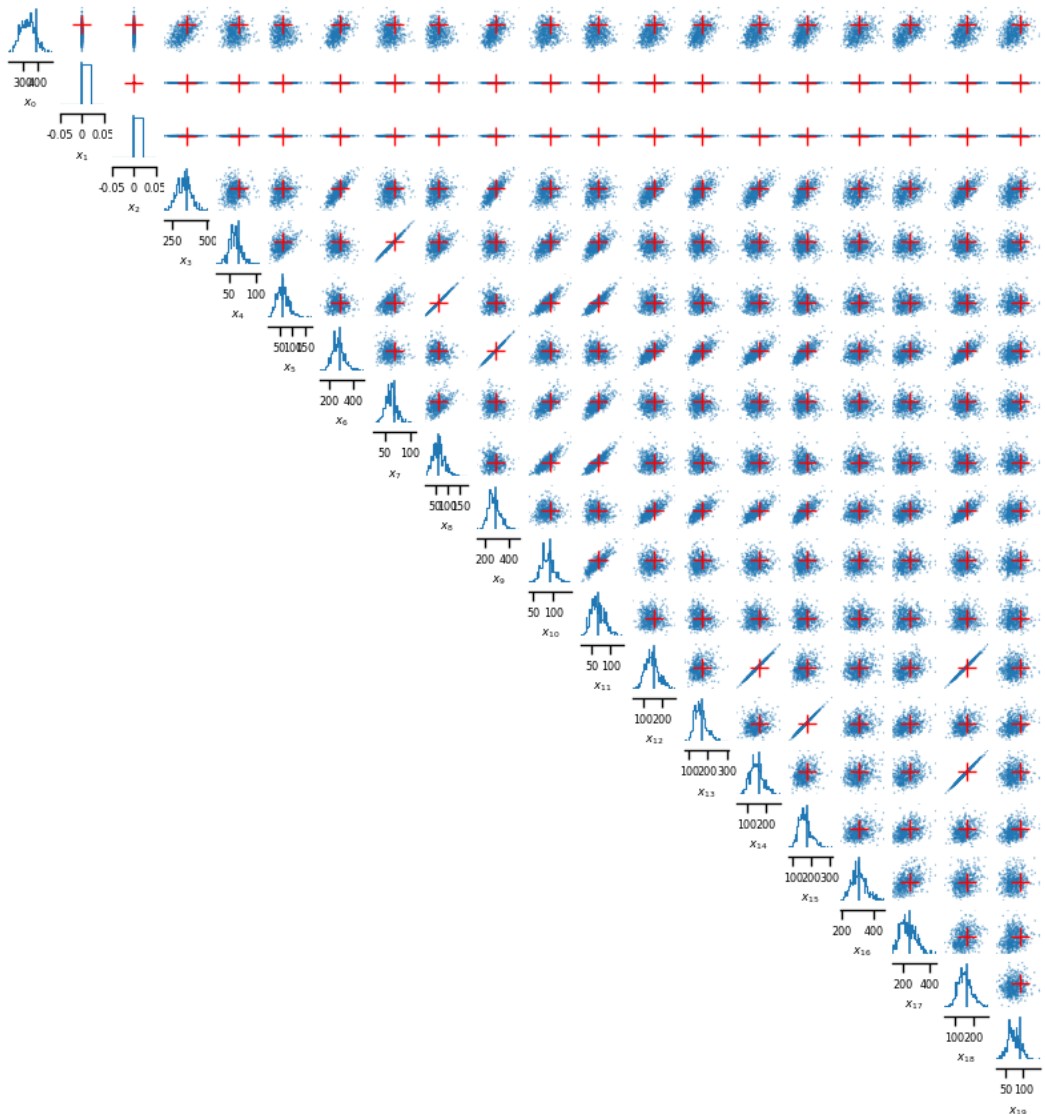

Figure 13: Pairwise density plots of a 20-dimensional slice of the "empirical likelihood" under the final ASNPE posterior on the *Prior I, Hours 5:00-6:00, Congestion level A* scenario. Figure 12 shows the approximate posterior $q_\phi(\theta|x_o)$, whereas here we draw samples $\theta_i \sim q_\phi(\theta|x_o)$ and feed them back through the simulator $\{\theta_i\}_{1:N} \to p(x|\theta)$ to visualize the resulting data space. The target observational data point $x_o$ is shown on top these pair plots as a red "plus", which provides a visual anchor for how well calibrated the posterior is around the observational data point of interest.

