# OpenReview forum: "Active Sequential Posterior Estimation for Sample-Efficient Simulation-Based Inference"
_NeurIPS.cc/2024/Conference — NeurIPS 2024 poster_

### Official Review · Reviewer_jL6p · 2024-07-02

**Soundness:** 3
**Presentation:** 3
**Contribution:** 3
**Rating:** 7
**Confidence:** 3

**Summary:**

This work introduces a new way to do active learning in simulation-based inference. This algorithm allows for the sampling of data points not only in regions of interest but also in regions that lead to high information gain. They then apply this algorithm to the problem of urban travel demand calibration, which consists of estimating the expected travel demand between different points of interest.

**Strengths:**

### Originality
* I'm not aware of other pieces of work introducing active learning in sbi the same way as done in this paper.
* I'm unfamiliar with the urban travel demand calibration literature, but I am also unaware of a similar piece of work.

### Quality
* The idea of targeting samples of high epistemic uncertainty in SBI is sound.
* The different approximations made give rise to a practical algorithm.
* The experiments suggest that the active learning scheme allows the reduction of the number of samples required from the simulation for the task of urban travel demand calibration.

### Clarity
* Overall, the paper is clearly written. The method and motivation are well described, step by step and the urban travel demand calibration task is also well described.

### Significance
* Introducing active learning in sbi is of high significance.
* I'm unfamiliar with the urban travel demand calibration literature but I would trust the authors on the fact that this is an important topic.

**Weaknesses:**

### Originality
* I have no concerns regarding originality.

### Quality
* I find that Equation 4 is not well motivated. Why are the terms coming from equation 3 and the proposal prior simply multiplied ? I would have expected an hyperparameter controlling the weight assigned to both those terms but there is not, and this choice of simply multiplying the terms seems arbitrary.
* In my understanding, optimizing equation 4 can be very computationally costly as it requires evaluating the Bayesian neural network on many samples and each evaluation of Bayesian neural network consist itself in the evaluation of several neural networks. I find that this limitation is not well stated in the paper.
* RMNSE is a strange metric to use for posterior evaluation, in my opinion. This metric is designed to evaluate the quality of point estimation but not distributions. In particular, it falls short when the posterior is multi-modal; the metric would prefer an unimodal approximation in the middle to the correct multimodal distribution.

### Clarity
* The paper is a mix of methodological and applicative paper and does not have a clear scope. While the title suggests an applicative paper, the authors introduce a new algorithm that is valuable in all generality for the field of simulation-based inference. The experiments are, however, limited to urban travel demand calibration and hence do not demonstrate the this method is effective for other problems.

### Significance
* I have no concerns regarding significance.

**Questions:**

Do you have results with a metric other than RMNSE that would not suffer from the same issues?

**Limitations:**

The limitations regarding the computational cost of the acquisition function are not mentioned.

---

> ### Author Rebuttal · Authors · 2024-08-07
>
> Thank you for your time and effort providing detailed feedback on our work. Please find our responses to your questions and comments below.
>
> Note: Where applicable, we prefix sections with `W-<x>`, `Q-<y>`, or `L-<z>` to reference itemized comments in Weaknesses, Questions, and Limitations, respectively, numbered by the order in which they were mentioned.
>
> **[W-1]** The motivation for Equation 4 is primarily discussed in Section 3.3 (lines 172-175), which describes how the $\theta^*$ that maximizes this notion of distributional uncertainty is not necessarily a likely parameter under the observational data $x_o$. Multiplying by the proposal prior, which captures the most up-to-date estimate of $p(\theta | x_o)$ at each round of the algorithm, allows us to re-weight this uncertainty by the approximate likelihoods of each parameter under $x_o$ and prioritize more likely $\theta$ during acquisition. This multiplication is primarily motivated by the desire to more closely align the effective likelihoods of parameters $\theta$ with the true proposal prior, which is ultimately present and corrected for in the NDE loss (see line 178).
>
> **[W-2, L-1]** Regarding computational cost of acquisition: Section 3.4 describes the mechanism by which Equation 4 can be optimized over a fixed set of proposal samples during each round. While we do indeed need to evaluate the NDE for several candidate parameters and many network realizations, this acquisition evaluation can be performed very efficiently as a batched inference step under the NDE model.
>
> Further, the additional computational overhead can be compared against SNPE (i.e., no acquisition function) for our primary traffic task, as the reported wallclock times (found in subfigures (b) for each of the calibration plots) provide the raw runtimes for each method when obtaining the 128 simulation samples for each setting. Empirically, this allows us to compare the total time spent by our algorithm (including the cost of optimizing the acquisition function), and we observe a negligible difference compared to SNPE over our explored horizons.
>
> **[W-3, Q-1]** You raise a valid point: RMSNE is indeed a non-standard metric for evaluating SBI methods in the literature, and is employed in this work almost entirely due to its prevalence in the OD calibration community. In our traffic calibration setting, there are few why it’s used:
>
> 1. We are ultimately interested in producing good point estimates of the true OD matrices,
> 2. RMSNE is a standard in the OD calibration space and allows us to compare with other baseline methods like SPSA and PC-SPSA, and
> 3. While we may benefit downstream from having a good distributional estimate under $x_o$ (as SBI methods generally attempt to provide), for our task, we are mostly interested in how approximating the posterior can improve our method's intermediate exploration of the parameter space in service of producing good point estimates.
>
> Nevertheless, we agree a systematic exploration of the final posterior accuracy under metrics better suited to capture distributional differences (not just point estimates) would be warranted in general. Please see our global rebuttal reply for more details here.
>
> **[W-4]** Please see our global rebuttal reply for details on additional empirical evaluation, including discussion around new simulation benchmarks and more representative metrics.

---

> > ### Comment · Reviewer_jL6p · 2024-08-12
> >
> > Thanks for your reply, I comment on it below.
> >
> > > The motivation for Equation 4 is primarily discussed in Section 3.3 (lines 172-175), which describes how the $\theta^*$ that maximizes this notion of distributional uncertainty is not necessarily a likely parameter under the observational data $x_o$. Multiplying by the proposal prior, which captures the most up-to-date estimate of $p(\theta | x_o)$ at each round of the algorithm, allows us to re-weight this uncertainty by the approximate likelihoods of each parameter under $x_o$ and prioritize more likely $\theta$ during acquisition. This multiplication is primarily motivated by the desire to more closely align the effective likelihoods of parameters $\theta$ with the true proposal prior, which is ultimately present and corrected for in the NDE loss (see line 178).
> >
> > I understand the need to introduce the $p(\theta | x_o)$ term. My comment was more that such acquisition
> > $$ \alpha(\tilde{\theta}, p(\phi|D)) = \tilde{p}(\tilde{\theta}) (E_{\phi' \sim \phi | D}[...])^\lambda$$
> > would also be perfectly valid for any $\lambda$ in my opinion and the choice of $\lambda$ = 1 seems arbitrary. The following acquisition would also be valid
> > $$ \alpha(\tilde{\theta}, p(\phi|D)) = (1-\lambda)\tilde{p}(\tilde{\theta}) + \lambda E_{\phi' \sim \phi | D}[...].$$
> > Therefore, I was wondering what was motivating your choice of acquisition function.
> >
> > > Regarding computational cost of acquisition: Section 3.4 describes the mechanism by which Equation 4 can be optimized over a fixed set of proposal samples during each round. While we do indeed need to evaluate the NDE for several candidate parameters and many network realizations, this acquisition evaluation can be performed very efficiently as a batched inference step under the NDE model.
> > Further, the additional computational overhead can be compared against SNPE (i.e., no acquisition function) for our primary traffic task, as the reported wallclock times (found in subfigures (b) for each of the calibration plots) provide the raw runtimes for each method when obtaining the 128 simulation samples for each setting. Empirically, this allows us to compare the total time spent by our algorithm (including the cost of optimizing the acquisition function), and we observe a negligible difference compared to SNPE over our explored horizons.
> >
> > Thanks for the clarification.
> >
> > >You raise a valid point: RMSNE is indeed a non-standard metric for evaluating SBI methods in the literature, and is employed in this work almost entirely due to its prevalence in the OD calibration community. In our traffic calibration setting, there are few why it’s used:
> > We are ultimately interested in producing good point estimates of the true OD matrices,
> > RMSNE is a standard in the OD calibration space and allows us to compare with other baseline methods like SPSA and PC-SPSA, and
> > While we may benefit downstream from having a good distributional estimate under $x_o$ (as SBI methods generally attempt to provide), for our task, we are mostly interested in how approximating the posterior can improve our method's intermediate exploration of the parameter space in service of producing good point estimates.
> > Nevertheless, we agree a systematic exploration of the final posterior accuracy under metrics better suited to capture distributional differences (not just point estimates) would be warranted in general. Please see our global rebuttal reply for more details here.
> > [W-4] Please see our global rebuttal reply for details on additional empirical evaluation, including discussion around new simulation benchmarks and more representative metrics.
> >
> > Thanks for the addition of new benchmarks with metrics better suited for simulation-based inference.
> >
> > Given the huge improvement in the evaluation of the method which was my main concern, I increased my score to 7.

---

> ### Author Response · Authors · 2024-08-14
>
> Thank you for the favorable score revision and again for your valuable feedback! It is very much appreciated. A few additional comments to the points raised:
>
> ***Regarding the formulation of the acquisition function***
>
> Thank you for clarifying. We agree, the family of functions
>
> $$\alpha(\tilde{\theta}, p(\phi, D)) = \tilde{p}(\tilde{\theta})(\mathbb{E}_{\phi^\prime\sim\phi |D}[\dots])^\lambda$$
>
> under parameter $\lambda$ constitutes valid choices for the acquisition function for any $\lambda$, facilitating different levels of emphasis on uncertainties at values of $\theta$ relative to their likelihoods under $\tilde{p}$. The choice to use $\lambda =1$ here is in part due to convenience as we did not set out to explore this parametric family explicitly, but it intuitively captures a desirable balance in the relationship between uncertainties and likelihoods of $\theta$.
>
> In particular, under level sets $\alpha(\cdot, p(\phi, D)) = z$ (where $p(\phi, D)$ is held constant), as likelihoods $\tilde{p}(\tilde{\theta})$ decrease by a factor of $n$, the average deviation between $p(\theta | x, D)$ and $p(\theta | x, \phi)$ need only increase by a factor of $\sqrt{n}$, i.e., changes in uncertainty are sub-linear in the likelihood ratio. With the introduction of $\lambda$, we’d see this factor generalize to $n^{1/(2\lambda)}$, and may need to take additional measures to balance the resulting sensitivity between the terms. We find that $\lambda =1$ is a natural choice that reasonably captures the desire to explore potentially unlikely parameters with high uncertainties without ignoring them (e.g., $\lambda \rightarrow 0$) or relying too heavily on them (e.g., $\lambda \rightarrow \infty$). We nevertheless find it intriguing to explore the impacts of $\lambda$ on the effectiveness of the acquisition function in practice, and will aim to incorporate this alongside our additional ablation tests in the final manuscript.
>
> Regarding the additive form, we find this slightly less intuitive, and potentially lacking some of the above-mentioned qualities. In particular, it only shifts the uncertainties for choices of $\theta$ rather than scaling them by their likelihood, meaning the relationship between terms is no longer dependent on a multiplicative factor. That is, under level sets of $\alpha$, absolute differences in likelihood $\tilde{p}(\tilde{\theta})$ need to be made up for by proportional absolute differences in uncertainty around $\tilde{\theta}$, which is slightly counter-intuitive (e.g., for any two $\theta_1,\theta_2$, the needed change in uncertainty is no longer determined by their likelihood ratio under $\tilde{p}$). Additionally, for $\lambda$ close to 1, $\theta$ with low likelihood may be too readily selected provided $\tilde{p}(\theta)$ is dominated by high uncertainty found outside of high-likelihood regions under $p(\theta | x_o)$.
>
> We welcome further discussion if there are any outstanding questions or concerns. Thank you again!

---

### Official Review · Reviewer_s4rF · 2024-07-02

**Soundness:** 2
**Presentation:** 3
**Contribution:** 2
**Rating:** 6
**Confidence:** 4

**Summary:**

The paper considers the problem of efficient neural density estimation for simulation-based inference, in settings where we want to estimate the parameters of a model from which we can draw samples but cannot define a likelihood function. Essentially, the main contribution of the paper is to integrate an active learning approach for exploring the parameter space efficiently, for which they define a measure by which they decide which parameter values should be "explored" next, i.e. used to generate more data to train the posterior. The approach is used for the Origin-Destination Matrix estimation problem in traffic simulation.

**Strengths:**

The main contributions of this paper are:

1) In case of simulation-based inference, an important question is how to explore the parameter space efficiently. This paper uses the concept of "active learning" for this purpose, by choosing the next few samples to "label" (by running simulations) using a measure
2) The paper brings on the concept of model uncertainty through Bayesian Neural Network, and marries it off with simulation-based inference
3) The paper considers elaborate experiments of origin-destination matrix calibration based on SUMO traffic simulations.

**Weaknesses:**

My general comment is that the paper has several useful and interesting ideas, but they have not been sufficiently explored or developed.

1) The experiments show improved sample complexity due to the "active learning". But if the true posterior distribution p(\theta|x_0) over parameter space is multimodal, then we may risk finding a suboptimal solution, especially if the prior is not suitable.
2) A measure is proposed to choose the next few samples to be "labelled", but we don't really get to understand why that measure should be used.
3) Although the general approach is quite generic, experiments are shown for only one task (OD calibration)

Minor comments:
Algo 1: I think you should initialize D(r) with D(r-1) outside the "for" loop for variable b, and inside the loop, it should be D(r)=D(r) U (\theta_b,x_b)
Fig 1 part b: internal text is illegible, should be expanded
Eq 4: p(\theta|x_0, \phi) should be p(\theta|x_0, \phi ')

**Questions:**

1) How to choose a good prior for the OD matrix? Can we use some insights based on additional knowledge about the locations?
2) What are the other endogenous and exogenous parameters used for this simulation?
3) The technique proposed is quite generic, but only one application is shown: OD matrix estimation for traffic simulation. Can you discuss other applications, maybe within the traffic simulation domain itself?
4) Is there any other way of dealing with the parameters \phi without defining a distribution over them? In other words, if we do not choose 'q' to be a Bayesian Neural Network, can we still use the active learning approach to choose candidate '\theta'?
5) Is Equation 3 basically the variance of \theta? Does the active learning basically choose those values at which \theta has maximum variance according its current posterior distribution?
6) When we are sampling candidate parameter values before applying the selection criteria, should we consider a distribution different from the current posterior? Choosing the current posterior essentially means we are "exploiting" the region of the parameter space where we already have observed some values - maybe we should "explore" the other parts of the parameter space using a different distribution?

**Limitations:**

.....

---

> ### Author Rebuttal · Authors · 2024-08-07
>
> Thank you for your time and effort providing detailed feedback on our work. Please find our responses to your questions and comments below.
>
> Note: Where applicable, we prefix sections with `W-<x>`, `Q-<y>`, or `L-<z>` to reference itemized comments in Weaknesses, Questions, and Limitations, respectively, numbered by the order in which they were mentioned.
>
> **[W-1]** This is true, although SBI/Bayesian methods in general are sensitive to a choice of prior, and this is not a limitation exclusive to our method. Further, the active learning mechanism does not change the limiting behavior of the method insofar as its relation to APT [1], as discussed in Section 3.4 (i.e., we still have $q_\phi(\theta |x)\rightarrow p(\theta|x)$ as $N \rightarrow \infty$).
>
> [1]: D. S. Greenberg, M. Nonnenmacher, and J. H. Macke. Automatic posterior transformation for likelihood-free inference, 2019.
>
> **[W-2]** The acquisition function proposed for selecting simulation samples is motivated across Sections 3.1, 3.2, and 3.3, with the principal motivation being to improve on traditional SBI methods by exploring the simulation parameter space in a more principled manner over the course of the multi-round inference procedure. In particular, we want to do so in a way that is maximally informative to our model of the posterior, which can yield compounding benefits in posterior accuracy under $x_o$ that are particularly important in settings with low simulation budgets and/or high simulation costs.
>
> **[W-3]** Please see our global rebuttal reply for details on additional empirical evaluation, including discussion around new simulation benchmarks and more representative metrics.
>
> **[W-4]** _Regarding syntax issues and small figure text_ Thank you for pointing these out, they will be corrected in our final manuscript.
>
> **[Q-2]** Many of these parameters are intrinsic to the SUMO simulator or the traffic network. For example, exogenous parameters include the route choice set and link attributes (e.g., free flow speeds, number of lanes). Endogenous variables include link and path traffic statistics (travel times, flows, speeds), and departure times.
>
> **[Q-3]** Our proposed approach falls into the class of simulation-based inference methods, and is indeed intended to be generally applicable in settings where statistical inference under an arbitrary mechanistic model is needed. There are many scientific domains that define problems of this nature (e.g., inferring parameters of biological processes, calibrating physics models to real-world observations, etc). In the transportation space, SBI methods like the one proposed can be used not only for inferring network demand (OD matrices), but also for many other dynamics in urban settings, such as human mobility and public transportation. Urban designers and city planners can reason about effects of various proposed changes using simulation models, as well as leverage inverse models learned with SBI to better characterize parameters required to achieve desired outcomes.
>
> **[Q-4]** Evaluating the acquisition function as-is requires some means of inducing a distributional estimate over the parameters of the chosen model. BNNs and MC-dropout are flexible ways to achieve this in practice, but without this form our acquisition function cannot be used as intended. We are not otherwise aware of a clear way to sidestep this requirement without considering an entirely different acquisition function (and no such formulation exists in the SBI/BO literature, to the best of our knowledge).
>
> **[Q-5]** Equation 3 captures average differences in the assigned likelihoods between the posterior model ensemble ($p(\theta | x_o, D)$, see also the marginalization above line 157) and any particular model instantiation under the weight posterior. It could perhaps be likened to a notion of "distributional variance" around $\theta$ under the model weight posterior $p(\phi |D)$, but it is distinctly different from the variance of $\theta$ under the posterior $p(\theta | x_o)$.
>
> **[Q-6]** This is a great question, and very similar to the one that motivated this work. It is worth first noting that sequential SBI methods in general are focused on learning a particularly accurate picture of $p(\theta | x_o)$, i.e., the posterior under known observational data of interest. These methods explicitly condition on $x_o$ to produce each round’s proposal distribution, effectively “refocusing” the next round’s samples to reflect the model’s current understanding of $\theta$ that explain $x_o$. As a result, the posterior at a given round does not strictly correspond to regions previously explored in the parameter space (and so “exploitation” may be a bit misleading); rather, it captures the parameters expected to explain $x_o$ under the simulation model, given the simulation samples that have since been observed.
>
> In any case, there are a few challenges when thinking about using the alternative distribution you mention in this case:
>
> 1. How can we ensure convergence to the true posterior if sampling from a modified distribution?
> 2. (similarly) How can we ensure the resulting exploration is consistent with the prior?
>
> Our approach effectively implements your proposition by allowing the underlying model to "explore" regions of the parameter space it doesn't currently "understand" well, while remaining consistent with prior specifications and converging to the true posterior in the limit. Put another way, our method effectively produces the mentioned alternative "exploration distribution," but through a combination of components (the posterior model and a selection mechanism) that can be tractably mapped onto properties required for consistent Bayesian inference.

---

> > ### Comment · Reviewer_s4rF · 2024-08-08
> >
> > I thank the authors for their meticulous responses to all the questions. My main concern about the work was that it is focussing only on one very specific task, though the approach was quite generic (I see that other reviewers too had the same concern). I am satisfied by the author's decision to add a few benchmark tasks, and now I am leaning towards recommending acceptance.

---

> > > ### Author Response · Authors · 2024-08-10
> > >
> > > Thank you for the favorable score revision and again for your valuable feedback! It is very much appreciated, and we welcome any further questions or discussion.

---

### Official Review · Reviewer_S6Jd · 2024-07-10

**Soundness:** 2
**Presentation:** 1
**Contribution:** 2
**Rating:** 5
**Confidence:** 4

**Summary:**

The paper addresses the problem of computational cost, or alternatively, of sample efficiency in simulation-based inference (SBI). By employing an active learning scheme in sequential neural posterior estimators (SNPE), the proposed method achieves improved sample efficiency, which is paramount when dealing with expensive to evaluate scientific simulator-based models.

**Strengths:**

The paper has the following strengths:
* It addresses a relevant problem in SBI.
* The idea of using active learning to guide the sequential steps is interesting.
* The proposed method is applied to real-world traffic simulation problems.

**Weaknesses:**

* The methodology is not motivated properly, and at times seems quite ad hoc. The authors first mention the expected information gain (EIG), which is a principled criteria for active learning, but then they propose equation 2 as the criteria they use. It was not clear to me why this is a good choice, and how it related to EIG. The distance $\mathcal{D}$ in equation 2 became the quadratic loss in equation 3 without any justification/reasoning. I did not understand how the equation above line 157 was estimated. This seems like a critical quantity in the proposed method, and it would be nice if there is a discussion about it.

* Existing methods for SBI using neural density estimators do not typically provide uncertainty quantification for the weights and biases ($\phi$) of the neural network. There is very little emphasis placed on discussing this aspect. The authors mention Bayesian neural networks once in Section 2.2, and talk about MC-dropout for approximating the posterior of $\phi$. The paper lacks an in depth discussion and experiments about using them in the context of SBI (which has so far not been done to the best of my knowledge), and how they affect the proposed method (for instance, how to set the prior for the Bayesian neural network, sensitivity to the dropout rate, etc.).

* The paper includes experiments only on the traffic simulators. This wouldn't be an issue if this was an applied paper, or if the method was motivated by certain aspects of this problem which would generalise to other simulators as well. However, the method is presented in general, which is why I would expect a varied set of benchmark experiments. Given that a major portion of the references are from transportation research literature, perhaps this work, in its current state, would be more relevant for audience of that community.

* The clarity of writing can certainly be improved.

* The paper lacks discussion of the hyperparameters of the proposed method, its sensitivity to the choice of hyperparameters, and how to set them in practice.

* The literature on active learning for simulator-based models is not cited and discussed, see for instance the following papers. This makes it difficult to judge the technical novelty of the proposed method.
    * Gutmann and Corander (2016): https://arxiv.org/abs/1501.03291
    * Kleinegesse et al (2019): https://proceedings.mlr.press/v89/kleinegesse19a.html
    * Kleinegesse et al (2020): https://arxiv.org/abs/2003.09379

**Questions:**

* I did not understand the reasoning behind adjusting equation 3. Why multiplying with the prior makes sense?
* Line 140-141: not clear what 'cost' is referring to.
* Line 172-173: I did not understand what is being conveyed.
* Line 191-192 talk about recovering the true optimum of  $\alpha$ as $N$ tends to infinity. But in practice $N$ is much smaller. Does this statement hold even in the finite case?
* What is the computational overhead of the proposed method compared to SNPE?

Typos/grammatical errors:
* Line 96
* Line 127 (genetic instead of generic)
* Line 121-122 doesn't parse well

**Limitations:**

Please include a paragraph on the limitations of the proposed approach.

---

> ### Author Rebuttal · Authors · 2024-08-07
>
> Thank you for your time and effort providing detailed feedback on our work.
>
> Note: We prefix sections with `W-<x>`, `Q-<y>`, or `L-<z>` to reference itemized comments in Weaknesses, Questions, and Limitations, respectively.
>
> **[W-1]** The discussion around EIG was principally intended to help set the stage for desirable qualities of an acquisition function, despite the fact it is itself not directly possible to optimize in LFI settings due to its reliance on the likelihood $p(x|\theta,D)$. This drawback motivates an alternative formulation that captures these qualities, which begins with our introduction of a notion of distributional uncertainty (up to a divergence measure $\mathcal{D}$) in Equation 2. This is further connected to Equation 3 via the discussion in Section 3.3, which makes concrete an approach inspired by the general form in Eq. 2 and that seen in [1].
>
> Line 157 is an expansion of the marginalization involved over NDE model parameters $\phi$ in defining $p(\theta|x,D)$, and is included primarily to highlight the relationship between $p(\theta|x,D)$ and $p(\theta|x, \phi)$. More exposition will be added to our final manuscript to better contextualize this form.
>
> [1]: K. Kandasamy, J. Schneider, and B. Poczos. Bayesian active learning for posterior estimation. (IJCAI ’15)
>
> **[W-3]** CI-1 Please see our global rebuttal reply for details on additional empirical evaluation, including discussion around new simulation benchmarks and more representative metrics.
>
> **[W-4]** Appendix C.1.1 provides the hyperparameters used by our algorithm across explored experimental settings, as well as our NDE model’s structure, dropout rate, etc. We agree that a more principled analysis of hyperparameter sensitivity of heuristics for setting them in practice would be valuable. While performing such a study is prohibitive on our primary task due to computational constraints, the effects of the algorithm hyperparameters (e.g., $R$, $N$, $B$) and NDE hyperparameters (e.g., network structure, dropout rate, etc) will be explored on the smaller scale SBI benchmarks discussed above and reported in our final manuscript.
>
> **[W-5]** These are valuable resources and indeed provide useful context for leveraging implicit models in Bayesian optimization contexts. While we believe mentioning these works in our literature review is warranted, there are a few reasons why they might otherwise have been seen as indirectly relevant:
>
> - They primarily leverage GP surrogate models rather than models seen in more recent SBI literature (e.g., flow-based generative models) and those used in our work.
> - Our efforts are focused on formulations that only require a direct posterior approximation, rather than assuming surrogate likelihoods or likelihood ratios are also available.
> - They primarily focus on experimental design contexts, which embrace a slightly different set of assumptions, formulations, and intended applications.
>
> **[Q-1]** The adjustment to Eq. 3 is motivated in Section 3.3 (lines 172-175), which describes how the $\theta^*$ that maximizes this notion of distributional uncertainty is not necessarily a likely parameter under the observational data $x_o$. Multiplying by the proposal prior, which captures the most up-to-date estimate of $p(\theta | x_o)$ at each round of the algorithm, allows us to re-weight this uncertainty by the approximate likelihoods of each parameter under $x_o$ and prioritize more likely $\theta$ during acquisition. See also [Q-3] below.
>
> **[Q-2]** “Cost” here is referring to the time or compute resources spent evaluating the simulator under the mentioned parameter. We agree this should be more clearly stated.
>
> **[Q-3]** Eq. 3 captures a means of quantifying uncertainty over the likelihood values assigned to $\theta$ under the NDE and its various realizations under $\phi\sim p(\phi|D)$. This does not capture the desire for $\theta$ to also be likely under the posterior $p(\theta|x_o)$, however. Our approach, along with many sequential SBI methods, is primarily concerned with learning an accurate view of the $p(\tilde{\theta}|x_o)$ under observational data $x_o$. Lines 172-173 are simply connecting this larger goal to the implications of optimizing Eq. 3, suggesting $\theta^*$ may be of low value if it’s unlikely under the observational data $x_o$. This motivates the adjustment to Eq. 3 that appears in Eq. 4 (addressed in [Q-1]).
>
> **[Q-4]** Lines 191-192 convey that the optimum of $\alpha$ (call it $\alpha^*$) is captured in a sample $X_n$ of size $n$ (drawn over $\alpha$'s domain) in the limit $n \rightarrow \infty$. This could be expanded to include the relevant implications, namely,
>
> $$p(\alpha^* \in X_n) \rightarrow 1, n \rightarrow \infty$$
>
> or
>
> $$\text{argmax}_{\theta\in X_n} \alpha(\theta, p(\phi | D)) \rightarrow \alpha^*, n \rightarrow \infty$$
>
> This does not suggest that there exists a finite sample of size $n$ within which $\alpha^*$ is present, and instead only captures the limiting behavior for increasingly large sample sizes. In practice, we leverage this consistency with $\alpha^*$ given this limiting behavior.
>
> **[Q-5]** The primary difference between ASNPE and SNPE in terms of computational overhead is the optimization of the acquisition function in Eq. 4. As is mentioned in Section 3.4, however, the acquisition function is computed over a fixed size parameter batch and NDE realizations $\phi\sim p(\phi|D)$ (with specific hyperparameters reported in Appendix C.1.1). This can be performed efficiently as a batched inference step under the NDE model, and typically yields a negligible difference in terms of raw runtimes between the two methods. For the traffic case study we explore in the paper, the raw runtimes of both methods (found in subfigures (b) for each of the calibration plots) allow us to compare the total time spent by our algorithm (including acquisition evaluation), and we observe a negligible difference compared to SNPE over our explored horizons.

---

> > ### Comment · Reviewer_S6Jd · 2024-08-12
> >
> > I thank the authors for the detailed responses. I am happy to see the inclusion of some benchmark examples that definitely improve the paper.
> > * **W-1:** I am still unclear about the role of $\mathcal{D}$ here, as it is taken to be the squared loss (if I understand correctly), and not any divergence/distance metric on the space of distributions.
> > * **W-2:** I do not see any response to this comment.
> > * **W-4:** This is exactly why it makes sense to have some simpler benchmarking experiments when proposing new methods, so that we can test the performance by varying different settings and hyperparameters in a "sandbox" environment, and develop intuition about the limits and workings of the method. I hope such detailed analyses are included in the future versions of the manuscript.
> > * **W-5:** Thank you. The points you mentioned are exactly what needs to be in the paper to put the proposed method in context with relevant literature.
> > * **Q-1:** I feel like the problem is with the sentence in lines 172-173. It is perhaps too dense for readers to understand (seems like another reviewer had similar confusion like me). A bit of explanation might help the readers.
> >
> > Given that the paper has improved quite a bit, I am happy to raise my score. All the best!

---

> > > ### Author Response · Authors · 2024-08-13
> > >
> > > Thank you for the favorable score revision and again for your valuable feedback! It is very much appreciated. A few additional comments to the points you raised:
> > >
> > > - **[W-1]** Equation 2 is mostly used as a general form that highlights the central idea of computing the expected difference between the marginal posterior and individual realizations of the NDE, for $\phi\sim p(\phi | D)$. This gives us a functional form dependent on specific choices of $x$, but not specific $\theta$, as it leverages the entire distributional form of the posterior (i.e., $\mathcal{H}_{\mathcal{D}}(x)$ is a scalar). Equation 3, however, uses the exact likelihood values at particular choices of $\theta$, and while loosely borrowing the approach from Eq. 2, it is not a downstream result of selecting any particular divergence measure. That is, it provides a notion of uncertainty induced by $p(\phi | D)$ that can be measured for specific parameter candidates $\theta$, and thus a divergence measure isn’t directly applicable here.
> > >
> > > - **[W-2]** _(Apologies, we ran out of space in our initial rebuttal)_ While quantifying uncertainty over NDE parameters is indeed uncommon in SBI settings, we viewed this mostly as a stepping stone to enable the use of our proposed acquisition function. While we provide some context in Appendix C.1.1, a more principled analysis of model hyperparameters is planned for the final manuscript as detailed in [W-4]. Additionally, while there is some exposition in Section 3.4 that highlights possible choices of NDE that both align with typical options in SBI and provide parameter uncertainties, we agree this should be further expanded upon, and will incorporate this in our final version.
> > >
> > > - **[W-4]** We agree, and will be sure to include these results, analyses, and ablations as laid out in our global rebuttal.
> > >
> > > - **[W-5]** We also think a discussion relating these works to our paper would help better position our method, and will include this in our updated manuscript.
> > >
> > > - **[Q-1]** Understood, thank you for making note of this; we’ll revise the wording here.
> > >
> > > We welcome further discussion if there are any outstanding questions or concerns. Thank you again!

---

> > > > ### Comment · Reviewer_S6Jd · 2024-08-13
> > > >
> > > > Thank you for your response. I greatly appreciate this healthy discussion and am happy to see that the authors are taking all the reviewers' comments constructively and actively working towards improving their manuscript.
> > > >
> > > > This discussion regarding [W-1] is emblematic of why I found the method description confusing and ad hoc while reading. Introducing notation that is not really used (like $\mathcal{D}$) just disrupts the flow while reading. I would encourage the authors to take such clarifying questions from reviewers as signals for things that are unclear in the paper, so as to improve the exposition a bit. This would greatly improve their paper in my opinion (apart from all the other improvements they have already made). Thank you.

---

### Official Review · Reviewer_JwDn · 2024-07-11

**Soundness:** 2
**Presentation:** 2
**Contribution:** 2
**Rating:** 4
**Confidence:** 3

**Summary:**

This paper proposes an approach to performing neural simulation-based inference – specifically, sequential neural posterior estimation – in a simulation-efficient manner for complex and expensive simulation models. The idea is to use active sampling to sequentially generate datapoints from the simulator and to train the neural density estimator using batches of datapoints generated this way. The authors test their approach experimentally on an origin-destination estimation problem in an urban mobility simulator, comparing against other common simulation-based Bayesian inference methods ("vanilla" SNPE and ABC) and against two alternatives that are commonly used in the urban mobility modelling literature ((PC-)SPSA).

**Strengths:**

_Originality_

As far as I am aware, active sampling via an explicit acquisition function as described in this paper has not been explored yet in the neural SBI literature, although some less formal active sampling already occurs in sequential/round-based training of neural SBI methods.

_Quality_

The method looks overall reasonably sound and some comparison against multiple alternatives is presented for the case of a complex urban mobility model.

_Clarity_

The paper was mostly clear in my reading, although there were one or two points that were a little unclear to me that I will detail below.

_Significance_

Overall I think this paper has (and approaches to making simulation-based inference procedures as simulation-efficient as possible, more generally, have) the potential to be significant and will assist practitioners in real-world settings to make use of simulation models effectively.

**Weaknesses:**

The two main weaknesses in my reading are:

- Sorry if I missed something, but clarity was lacking in one key respect for me, namely what exactly the relationship was between $q_{\phi}(\theta \mid x)$, $p(\theta \mid x, \phi)$, $\tilde{q}_{x,\phi}(\theta)$ on Line 178, and `q_{\phi,x}(\theta)` (I had to do this last one `in this format` rather than in $math\ mode$, sorry, because it wasn't rendering properly in $math\ mode$ and I couldn't figure out why) on Line 178. Is `q_{\phi}(\theta \mid x)` (now $math\ mode$ isn't working here either...) the same as `q_{\phi,x}(\theta)`? And `\tilde{q}_{x,\phi}(\theta)` the same as `p(\theta \mid x, \phi)`? If so can you use consistent notation throughout for clarity? And if not could you please clarify what the difference between all of these guys are?

- The empirical evaluation was quite weak in my opinion. The authors present a new approach to training SNPE models and it's great that they've applied it to a complex simulator to get some idea of whether it works on real practical simulators of interest, but I think it's also important to test a new method on some simpler benchmark models for which a good (even if approximate) ground-truth posterior can be obtained. This will allow the authors to properly test whether their method produces good posteriors, which isn't really tested at the moment by looking purely at RMSNE values. In general I would like to see something that demonstrates how well the proposed Bayesian pipeline can actually estimate the full posterior distribution, since this is ultimately what it's trying to do. I think the paper needs revision to include such experiments, even if the detail on this is mostly relegated to the appendix.

A couple of less substantial weaknesses are:

-  Perhaps the literature review could be a bit more extensive/comprehensive. For example, reference [1] below is a(n admittedly very) recent paper on origin-destination matrix estimation in mobility models, and while it might not be necessary or possible to compare against, it would perhaps be worthwhile including in the literature review as a SOTA method for estimating OD matrices when evaluating the likelihood is extremely expensive. Further, reference [2] below also considers the problem of efficiently performing SBI for complex and expensive simulators, albeit with a focus on obviating the task of learning summary statistics so a slightly different focus.

- Some proof-reading and fixing of formatting issues is needed. For example: Line 268 is missing a word ("...PC-SPSA is an effective extension _that over_ parameters in a lower-dimensional subspace...") as is Line 294 ("...ASNPE is outperformed by _PC-SPSA two_ of our explored settings..."). Also the lines following Lines 141 and 156 have no numbers, not sure what's happened there (not a big deal currently but might cause problems down the line).

[1] _Zachos, I., Damoulas, T., & Girolami, M. (2024). Table inference for combinatorial origin‐destination choices in agent‐based population synthesis. Stat, 13(1), e656._

[2] _Dyer, J., Cannon, P. W., & Schmon, S. M. (2022, May). Amortised likelihood-free inference for expensive time-series simulators with signatured ratio estimation. In International Conference on Artificial Intelligence and Statistics (pp. 11131-11144). PMLR._

**Questions:**

See above ^ and thanks in advance for your responses!

**Limitations:**

I do not see that this work especially threatens any negative societal impact. The main limitations of the work that I see are already addressed in the Weaknesses section above.

---

> ### Author Rebuttal · Authors · 2024-08-07
>
> Thank you for your time and effort providing detailed feedback on our work. Please find our responses to your questions and comments below.
>
> Note: Where applicable, we prefix sections with `W-<x>`, `Q-<y>`, or `L-<z>` to reference itemized comments in Weaknesses, Questions, and Limitations, respectively, numbered by the order in which they were mentioned.
>
> **[W-1]** There is indeed some overlap in notation, and the differences between these forms are mostly context-dependent:
>
> - $q_{\phi}(\theta | x)$: the approximate posterior NDE model, with parameters $\phi$
> - $p(\theta | x, \phi)$: a concrete posterior density produced by a particular setting of NDE weights $\phi$
> - $\tilde{q}_{x, \phi}(\theta)$: the approximate proposal prior
> - $q_{\phi, x}(\theta)$: same as $q_{\phi}(\theta | x)$, but with syntax adjusted to mirror the key result being used from [1] at line 178.
>
> In particular, the form used on line 178 was constructed to resemble the setup of a key result we leverage from [1]. However, we agree that this may ultimately have made the connection to the notation elsewhere in our methodology unclear, and will revise this in our final manuscript (by both introducing each term if using a different form, and simplifying where applicable).
>
> [1] D. S. Greenberg, M. Nonnenmacher, and J. H. Macke. Automatic posterior transformation for likelihood-free inference, 2019.
>
> **[W-2]** Please see our global rebuttal reply for details on additional empirical evaluation, including discussion around new simulation benchmarks and more representative metrics.
>
> **[W-3]** _Regarding paper [1]_: We distinguish between the following two problems in OD estimation:
>
> 1. General OD estimation problems (also known as travel demand estimation problems), where the outputs are stand-alone OD matrices that can be used for a variety of planning and operational network analysis, and
> 2. Model calibration problems, where the goal is to calibrate (or estimate) the inputs (such as the demand inputs specified as OD matrices) of a specific traffic simulation model and the output is a calibrated traffic simulator that can itself be used for analysis.
>
> Paper [1] addresses Problem (i), while our work addresses Problem (ii). We would like to stress that this distinction is an extremely important one. In particular, the main challenges of Problem (ii) are due to using an intricate (e.g., stochastic, non-differentiable, high compute cost) traffic simulator. This calls for likelihood-free methods, sample-efficient methods, and methods robust to simulator stochasticity. In contrast to this, Problem (i) can be formulated as a differentiable problem, which can be tackled with a likelihood-based approach, has little-to-no compute cost challenges, and no need for sample efficiency. This distinction is also discussed in [A].
>
> [A]: C. Osorio (2019) High-dimensional offline origin-destination (OD) demand calibration for stochastic traffic simulators of large-scale road networks. Transportation Research Part B: Methodological, Volume 124, Pages 18-43
>
> _Regarding paper [2]_: This work addresses the similar problem of performing likelihood-free inference under expensive simulators/low budgets. However, it focuses almost entirely on formulations suited for time-series data, and aims principally to jointly learn both summary statistics and a classifier for performing density ratio estimation. These are no doubt important settings in this space, but this particular work ultimately differs in several significant ways form our paper (e.g., we use a single observational data point, work exclusively with direct posterior approximations, require a means of inducing uncertainty over NDE parameters, etc).
>
> **[W-4]** Thank you for pointing these out, lines 268 and 294 are indeed missing words and will be corrected in our final manuscript. We also noticed the missing line numbers in several places (appears to be an odd formatting bug) and will ensure these are fixed.

---

> > ### Comment · Reviewer_JwDn · 2024-08-08
> >
> > Thanks for your rebuttal.
> >
> > **[W-1]**: Thanks for explaining. I do think consistency in notation (e.g., picking either bullet point 1 or bullet point 4 to use throughout) will be important for the paper's clarity. I also do not personally see why it's important to distinguish between bullet point 1 and bullet point 2. Would it not perhaps be clearer to use $\lbrace{q_{\phi} \mid \phi \in \Phi\rbrace}$ or something to refer to the NDE model and $q_{\phi}$ for a particular instantiation (i.e. the NDE at parameter values $\phi$)?
> >
> > **[W-2]**: Thanks for taking the time to prepare these additional experiments, I think this is valuable to include in the updated paper.
> >
> > **[W-3]**: Thanks for considering how your contribution relates to these papers. I think the differences you raise make sense, and would be important to discuss in the revision to more thoroughly contextualise your work.
> >
> > With these improvements in place I would be happy to raise my score.

---

> > > ### Author Response · Authors · 2024-08-12
> > > **Follow-up on Planned Revisions**
> > >
> > > Thank you again for your feedback, we wanted to briefly follow up on our planned revisions mentioned above. Please let us know if there are any further questions we can address; we’d be happy to continue the discussion if there are outstanding concerns or revision suggestions. Thank you!

---

> ### Author Response · Authors · 2024-08-10
>
> Thank you for your consideration regarding the score, as well as the additional feedback. It is very much appreciated!
>
> **[W-1]** We agree: the use of $q_{\phi, x}(\theta)$ (bullet #4) is inconsistent and should be replaced with $q_{\phi}(\theta | x)$ (bullet #1). We will make this change in our final version.
>
> Regarding $q_{\phi}(\theta | x)$, or $q_\phi$, we generally use this to indicate that $q$ is a family of models parameterized by $\phi$. $p(\theta|x, \phi)$ is then used in contexts when $\phi$ is concrete (e.g., in an expectation like that of Eq. 2, or in the marginalization above line 157), and parallels the generic form we see with the marginal posterior $p(\theta | x, D)$. However, as you point out, this distinction isn’t particularly important (or at least the use of two separate forms isn't necessary), and we agree simply using $q_\phi(\theta |x)$ directly is better in these contexts given it's clear $\phi$ is concrete (i.e., it doesn't clash with the broader notion that $q_\phi$ can refer holistically to the model family). We will make these substitutions in our final version.
>
> **[W-3]** We also think this discussion is valuable, and will be sure to include it in the literature review of our final manuscript.
>
> ---
>
> In summary, **W-1**, **W-2** (addressed in global rebuttal), and **W-3** will each be incorporated in our final version. Thank you again for your consideration, and we welcome any further questions or discussion.

---

> ### Author Response · Authors · 2024-08-13
> **Request for Final Score Revision**
>
> Given you indicated a willingness to revise your score, we wanted to kindly ask if you would look over our reply, which includes our plan to incorporate your suggestions (_W-1_, _W-2_, and _W-3_). If these changes meet your expectations and warrant the aforementioned score revision, doing so prior to today’s deadline would be greatly appreciated. Thank you again for your helpful feedback during this process!

---

### Official Review · Reviewer_GNuF · 2024-07-12

**Soundness:** 3
**Presentation:** 4
**Contribution:** 3
**Rating:** 6
**Confidence:** 3

**Summary:**

A recent class of methods that have shown to perform well at simulation-based inference are based on modeling the posterior density as a neural network, using mixture density networks, normalizing flows, or other popular architectures. With enough data, these methods tend to produce salient estimates of the posterior density on parameters conditional on data (or summary statistics). With any simulation-based inference method, proposing parameters efficiently is a relevant concern because the posterior can have most of its mass in a tiny region of the support of the prior. Standard SNPE constructs sequential proposal distributions from the current posterior approximation and uses importance reweighting, which is a simple heuristic but not optimal. The main contribution in this paper is an active learning method which filters potential parameter values by ones that are sufficiently large in terms of an acquisition function, such as expected information gain. The authors come up with a reasonable acquisition function that can be approximated via Monte Carlo. The ASNPE algorithm is then applied to an origin-destination calibration problem, and performs well compared to competing approaches under a budget of simulator evaluations.

**Strengths:**

The paper is clear about the contributions, and contextualizes them well relative to the previous work. Active learning is a desired traits in simulation-based inference methods, where simulators can be costly. The active learning component that is introduced in this work seems like a clear improvement over the standard heuristic employed by SNPE.

**Weaknesses:**

I would be interested to see more experiments/applications than just the Bayesian O-D calibration example. Particularly, I'd like to see how this method compares to competing approaches on a standard "benchmark" in likelihood-free/simulation-based inference, i.e. Lotka-Volterra, a queue model, Heston model, a Gaussian toy model as in SNL, etc. This particular simulator model seems to have an extremely high-dimensional parameter space relative to the budget of simulations used for the experiments, and I am curious if it is this particular setting in which this method shines compared to others. I.e. the benefit is clear when the simulation budget is 128, but if it is on the order of 10^4 or 10^5, does it vanish?

**Questions:**

1.) I am curious how much the computational cost increases when this methodology is used compared to the simple sequential updating in SNPE. Of course, when the simulator is extremely expensive, the cost of evaluating the acquisition function will be small in comparison, but would be it still be recommended if the simulator is not too onerous.

2.) On a similar note, I would be interested for the authors to carve out and add a little exposition describing exactly the class of SBI problems that they posit would have the largest boon from this active learning approach when compared to the standard sequential updating in SNPE. What types of problems are most rewarded by the acquisition function filtering step, and on which class of problems is it less rewarding?

3.) I may have missed it, but is it obvious that q_\phi still converges to the true posterior in the presence of the filtering step? In the presence of the filtering step, the proposal is no longer just \tilde{p}(\theta), but instead it is a transformed distribution essentially weighting \tilde{p}(\theta) by the acquisition function. Shouldn't this need to be considered when updating the NDE in order to ensure q_\phi really converges to the posterior?

**Limitations:**

The authors provided an honest assessment of the performance of their method in comparison with other methods in the experiments section. I believe that potential negative societal impact is not a concern here. As I mentioned earlier, I would be interested to know if the authors believe there are limitations in the context of which SBI problems their method is not as well-suited for.

---

> ### Author Rebuttal · Authors · 2024-08-07
>
> Thank you for your time and effort providing detailed feedback on our work. Please find our responses to your questions and comments below.
>
> Note: Where applicable, we prefix sections with `W-<x>`, `Q-<y>`, or `L-<z>` to reference itemized comments in Weaknesses, Questions, and Limitations, respectively, numbered by the order in which they were mentioned.
>
> **[W-1]** Please see our global rebuttal reply for details on additional empirical evaluation.
>
> ***Regarding vanishing benefit***: While we haven’t been able to experiment with large sample sizes (e.g., of the order 10^4-10^5) in our high-dimensional traffic setting due to computational constraints in our academic environment, we expect ASNPE’s advantage over SNPE to diminish as we collect increasingly large amounts of simulation samples. This is primarily due to the fact ASNPE and SNPE converge to the true posterior in the limit under samples from the simulation model, and both take similar steps to improve the accuracy of the posterior approximation under the observational data of interest. Precisely at what simulation budget we might expect the difference between methods to become negligible, however, is difficult to anticipate, and would require a more principled empirical study.
>
> To this end, we will explore this question alongside the experimental results on the common SBI benchmarks mentioned above in our final manuscript. Given the smaller scale simulation models, we can better characterize the relationship between the dimensionality of the parameter space and the behavior of ASNPE and SNPE over longer simulation horizons.
>
> **[Q-1, Q-2, L-1]** You raise a valid concern: the cost required for evaluating the acquisition function may not always be worthwhile when the simulation model is very cheap to evaluate. In these cases, more data may be more valuable than a principled exploration of the parameter space, and the time ASNPE spends on active learning may be better spent collecting additional simulation samples.
>
> Precisely characterizing the kinds of problems where we expect the cost for acquisition to be worthwhile is difficult, but in general we posit our method will provide the most benefit in complex settings (e.g., stochastic, non-trivial, large parameter/observation spaces) where simulation samples are limited (due either to slow simulation models, limited computational resources, or both). Our primary traffic case study is a problem very much characterized by these qualities, which is at least in part why we elected to focus most of our efforts on this single problem. In these cases, ASNPE may exhibit large benefits over traditional methods given its more principled exploration of the parameter space, and with sequential inference, early improvements can compound heavily over time (i.e., informative samples produce better proposal distributions, from which the next round’s samples are drawn, and so on).
>
> Regarding ASNPE’s relative computational overhead, note that acquisition evaluation can be performed very efficiently as a batched inference step under the NDE model, and typically yields a negligible difference in terms of raw runtimes between the two methods. For our primary traffic task, the reported wallclock times (found in subfigures (b) for each of the calibration plots) provide the raw runtimes for each method when obtaining the 128 simulation samples for each setting. Empirically, this allows us to compare the total time spent by our algorithm (including acquisition evaluation), and we observe a negligible difference compared to SNPE over our explored horizons.
>
> **[Q-3]** The convergence of $q_\phi$ to the true posterior is discussed in Section 3.4. In particular, we leverage a result from APT [1] (also known as SNPE-C), which states that training our NDE model parameters via maximum likelihood $\min_\phi \tilde{\mathcal{L}}$, where
>
> $$\tilde{\mathcal{L}}(\phi) = -\sum_{i=1}^N \log \tilde{q}_{x,\phi}(\theta_i)$$
>
> and
>
> $$ \tilde{q}_{x,\phi}(\theta) = q_{\phi}(\theta)\frac{\tilde{p}(\theta)}{p(\theta)}\frac{1}{Z(x,\phi)} $$ (line 178)
>
> implies $q_\phi(\theta|x) \rightarrow p(\theta|x)$ as $N \rightarrow \infty$ (along with the proposal posterior $\tilde{q}_{x,\phi}(\theta)\rightarrow \tilde{p}(\theta|x)$). This setup allows us to incorporate the true proposal prior $\tilde{p}(\theta)$ in the training loss without additional explicit corrective terms. Further, this limiting behavior holds even when optimizing $\tilde{\mathcal{L}}$ over samples drawn from the transformed distribution under the acquisition filter, as it shares its support with the proposal prior (and the same can be said for any distribution with this quality, in the limit as $N \rightarrow \infty$).
>
> [1] D. S. Greenberg, M. Nonnenmacher, and J. H. Macke. Automatic posterior transformation for likelihood-free inference, 2019.

---

> > ### Comment · Reviewer_GNuF · 2024-08-12
> >
> > Thanks for the rebuttal.
> >
> > [W-1]: I appreciate the addition of further evaluations on more standard SBI benchmarks, especially given the time constraint. My concerns were not necessarily about the lack of an in-depth evaluation of the advantage as a function of sample size, but rather that a sufficient amount of exposition should be given to elucidate to the reader the settings in which this methodology may be particularly helpful, and in which it may provide a significantly smaller benefit. I believe that the additional simulations that were described in the global rebuttal quell my concerns here.
> >
> > [Q-1, Q-2, L-1]: I do agree with the authors' perspective that the traffic simulation model showcases the benefits of this methodology well. I appreciate the clarification regarding the relative computational cost of the filtering step.
> >
> > [Q-3]: I also appreciate the clarification in this regard.
> >
> > I believe this work has value to practitioners and researchers of SBI and lean towards recommending acceptance.

---

> > > ### Author Response · Authors · 2024-08-13
> > >
> > > Thank you for recommending acceptance and again for your valuable feedback! It is very much appreciated, and we welcome any further questions or discussion.
> > >
> > > An additional note for **[W-1]**: Regarding exposition that better characterizes the problems for which our method is particularly ill/well-suited, the 2nd paragraph under **[Q-1, Q-2, L-1]** is likely the most relevant part of our original rebuttal to this point. We intend to expand on this further in our final version alongside the additional numerical results.

---

### Official Review · Reviewer_garc · 2024-07-12

**Soundness:** 3
**Presentation:** 3
**Contribution:** 3
**Rating:** 8
**Confidence:** 3

**Summary:**

The paper introduces ASNPE, a modification of sequential neural posterior estimation (SNPE) that uses active learning to determine the set of most informative simulation parameters. The method is benchmarked in a synthetic scenarios based on a real-world traffic network and outperforms domain-specific optimization schemes as well as simulation-based inference methods.

**Strengths:**

- The clear and concise communication throughout the paper makes it an enjoyable read.
- It features a simple and elegant approach to target a gap in current SNPE methods.
- Considering the empirical evaluations, a high-dimensional setting with real-world importance is chosen. ASNPE demonstrates a substantial improvement over both domain-specific optimization schemes and, at least for prior 2, SNPE. Again, the results are analyzed in a clear and informative way.
- Lastly, I appreciate the detailed description of the planned open source Python packages in Appendix D.

**Weaknesses:**

Major:
- While the reported experimental setup is exciting, the proposed method is tested only in a single setting. Since ASNPE seems to be a promising general improvement of SNPE-C, it would have substantially benefitted the work to demonstrate its usefulness in a wider range of experimental settings (e.g., by adapting tasks from [1] as a compact first experiment). I acknowledge that this is likely not feasible in the limited rebuttal period, but I believe it would make the paper much more informative for the broader SBI audience.

Minor:
- The paper directly jumps from theory to experimental setup and the results without mentioning the implementation of ASNPE, but this would be quite informative for the reader.
- Section 4.2 does not feature all benchmark comparisons - MC-ABC and SNPE suddenly appear in the results and I expected to find more information concerning MC-ABC at least in Appendix C. As a related minor comment, I suppose the benchmark uses APT / SNPE-C for SNPE, but it would be helpful to state this explicitly.
- Lastly, it would have been informative to see ablation studies (potentially in a computationally less demanding setting) or at least a theoretical discussion regarding the impact of the hyperparameters R, N, and B on the performance of ASNPE.

[1] Lueckmann, J. M., Boelts, J., Greenberg, D., Goncalves, P., & Macke, J. (2021). Benchmarking simulation-based inference. AISTATS.

**Questions:**

- In the training setup of SNPE and ASNPE, the number of simulations is not clear to me. The authors state that they perform R=3 round with selection size B=32, which should result in 96 simulations seen by the neural nets. How does this match the 128 sample simulation horizon?
- I (and maybe also future readers) would be interested in the authors opinion on the reasons for the clear gap between SNPE and ASNPE in the prior 1 but not prior 2 setting.

**Limitations:**

The explicit analysis of failure modes (which of course, are to be expected in any method comparison since no method always performs the best) is a big plus. As stated above, this does not incorporate a comparison between SNPE and ASNPE, which would be interesting for the wider SBI community.

---

> ### Author Rebuttal · Authors · 2024-08-07
>
> Thank you for your time and effort providing detailed feedback on our work. Please find our responses to your questions and comments below.
>
> Note: Where applicable, we prefix sections with `W-<x>`, `Q-<y>`, or `L-<z>` to reference itemized comments in Weaknesses, Questions, and Limitations, respectively, numbered by the order in which they were mentioned.
>
> **[W-1]** Please see our global rebuttal reply for details on additional empirical evaluation.
>
> **[W-2]** We agree this transition could be smoother. Section 3.4, however, sets up several practical considerations for how ASNPE can be implemented (e.g., efficiently approximating Eq. 4, training neural network-based NDEs with MC-Dropout, etc), and specific model details and hyperparameter settings are relegated to Appendix C.1.1 (primarily due to space constraints). In our final manuscript, we will incorporate a discussion on concrete implementation choices prior to the experimental setup, and direct readers to the Appendix for additional details.
>
> **[W-3]** The intended scope of Section 4.2 was to introduce methods specifically used in OD calibration literature (i.e., just SPSA and PC-SPSA), while we relied on prior context for the likelihood-free inference methods (i.e., SNPE and MC-ABC). However, as you point out, MC-ABC is not properly introduced, and the exact version of SNPE remains unclear (although it is indeed APT). In our final manuscript, we’ll broaden the scope of Section 4.2 to include all reported methods and properly introduce them.
>
> **[W-4]** This is a good point, and something we will incorporate alongside the smaller scale experiments run on benchmarks mentioned in [W-1]. As a quick theoretical discussion, however, the following reflects our observations on the large-scale setting we explore in the paper. Keeping the total number of simulation samples constant,
>
> - The number of rounds $R$ dictates how many times we update the proposal distribution over the course of the simulation horizon. Increasing this value can enable quicker feedback to the NDE, requiring fewer simulation samples before re-training the model. When the prior is well-calibrated and simulation samples are representative of the observational data, this can have a positive compounding effect that boosts the rate of convergence to the desired posterior. However, for larger R the resulting batch sizes are smaller and the NDE receives noisier updates, which can have the opposite effect and hurt early performance when the prior is poor.
> - The number of selected samples B per round is directly determined by R when the total number of simulations is held constant, and thus the above effects apply here.
> - The number of proposal samples N per round governs the size of the parameter candidate pool over which the acquisition function is evaluated. Increasing this value allows us to consider more potentially relevant candidates under $\tilde{p}(\theta)$, and can thus increase the quality of the resulting B-sized batch. Given the acquisition function can be evaluated over this pool very efficiently (i.e., as a batched inference step through the NDE model), one can practically scale this up arbitrarily to increase the sample coverage over the proposal support (but with decreasing marginal utility).
>
> **[Q-1]** Thank you for pointing this out. It should instead say R=4 (selection size B=32 is correct),  and will be corrected in our final version.
>
> **[Q-2, L-1]** Generally, prior I captures relatively high bias/low noise around the true OD parameter, while prior II is comparatively low bias/high noise. Our opinion as to why gaps between SNPE and ASNPE relies on a few observations:
>
> 1. Empirically, all methods under prior II benefit from its lower bias (seen very clearly in Figure 2) without suffering too greatly from the higher noise, indicating a particular sensitivity to the prior bias for this congestion setting and time of day.
> 2. ASNPE can identify useful simulation parameters more quickly than SNPE due its explicit acquisition mechanism, aligned with uncertainty in the NDE model.
>
> Given the sensitivity around the prior in this setting, ASNPE can have an edge over SNPE by making early strides to correct for the bias through exploration, and this can have a significant compounding effect for multi-round inference on such short simulation horizons. Put another way, ASNPE attempts to counter uncertainty in the NDE model by employing a more principled exploration of the parameter space than SNPE, and the effects of doing so are magnified under the higher bias under prior I (and less prominent under the less biased prior II).

---

> > ### Comment · Reviewer_garc · 2024-08-09
> >
> > Thank you for addressing my concerns. I especially value the additional experiments given the short time available.
> >
> > [W-1] I appreciate C2ST as an additional metric, but think it is important to note that the score is nearly 1 for all settings and methods and thus its informative value is very limited here. Acknowledging the papers focus on small simulation budgets, I still would recommend to additionally repeat the experiments with a bigger budget to better discriminate between SNPE and ASNPE regarding C2ST in a *final* version (not necessary during discussion period). If I understand the authors correctly, this is already planned with " 2) comparison between ASNPE and SNPE over long simulation horizons".
> >
> > The other responses greatly improve the clarity for me and would be informative in the final version. I believe the paper is valuable for the NeurIPS community and increased my score accordingly.

---

> > > ### Author Response · Authors · 2024-08-10
> > >
> > > Thank you for the favorable score revision and again for your valuable feedback! It is very much appreciated.
> > >
> > > **Regarding C2ST** We agree completely, and should emphasize that this metric in particular was empirically difficult to improve under the small sample sizes tested. This is mostly consistent with the C2ST values reported in [1] on the reference benchmark tasks, where C2ST is broadly quite close to 1.0 for small sample sizes (smallest reported being $10^3$). Indeed, we intend to scale up these trials and explore performance on larger simulation samples ($10^5$+) for the final version. _(Note that our initial attempts at doing so indicate the metric appears to improve significantly for even small multipliers on sample size. For instance, on the Bernoulli GLM task with a sample size five times as large (2500 simulation draws), C2ST is much more distinctive, with SNPE averaging ~$ 0.87$ and ASNPE ~$0.85$ across five trials.)_
> > >
> > > [1] Lueckmann, J. M., Boelts, J., Greenberg, D., Goncalves, P., & Macke, J. (2021). Benchmarking simulation-based inference. AISTATS.

---

### Author Rebuttal · Authors · 2024-08-07

We’d like to thank all reviewers for their time and effort in providing insightful feedback on our work. Please find our responses to your individual questions and comments in the rebuttal replies to each review.

Due to space constraints, we would like to address a common critique raised by many reviewers here in our global rebuttal regarding limited empirical evaluation. In summary, we agree that our work would benefit greatly from additional benchmark results, particularly for common simulation models in simulation-based inference (SBI) literature. To this end, we’ve evaluated our method on three tasks under three new metrics. Our hope is that these initial results provide additional depth when it comes to judging the empirical value of our contributions. These settings will be scaled up and explored in a more systematic fashion for presentation in our final manuscript.

### New benchmark tasks

We evaluated our method (ASNPE) and SNPE-C [1] on three common SBI tasks found in the literature (namely [2], which provides a suite of benchmarks across several SBI methods): **SLCP distractors**, **Bernoulli GLM**, and **Gaussian Mixture** (note that each corresponds to a reproducible task environment from [2]).

We further evaluated our method on these tasks using metrics beyond RMSNE that are better suited for capturing the accuracy of the approximate posterior, including _classifier 2-sample tests (C2ST)_, _maximum mean discrepancy (MMD)_, and _Kernelized Stein Discrepancy (KSD)_.

**Note**: Full details for each of these tasks and metrics can be found in [2].

We evaluated both methods over small sample horizons: 4 rounds at 125 samples per round, for a total of 500 simulation samples. Note that this is nearly four times larger than the sample sizes collected for the primary traffic setting explored in paper. For reference, 125 samples in our (non-parallelized) SUMO environment takes ~3 hours, whereas 125 samples from SLCP takes ~5 seconds on our hardware (see also Appendix C.1.1).

Note that these trials were repeated five times for each method, and the average score and standard deviation for each metric over these trials is reported in the tables below.

### Results

|       | MMD            | C2ST          | KSD           | L2            |
| :---  | :---           | :---          | :---          | :---          |
| SNPE  | 11.3820 ± 1.1043 | 0.9938 ± 0.0003 | 0.2901 ± 0.1217 | 5.9121 ± 0.3323 |
| ASNPE | **11.2673** ± 0.5796 | 0.9938 ± 0.0002 | **0.2558** ± 0.1331 | **5.6731** ± 0.1988 |
**Table 1: SLCP Distractors**

|       | MMD            | C2ST          | KSD           | L2            |
| :---  | :---           | :---          | :---          | :---          |
| SNPE  | 18.7130 ± 1.8315 | **0.9952** ± 0.0028 | 0.2496 ± 0.0579 | **37.2472** ± 3.4843 |
| ASNPE | **16.1893** ± 0.5625 | 0.9980 ± 0.0007 | **0.2340** ± 0.0467 | 39.4204 ± 0.2686 |
**Table 2: Bernoulli GLM**

|       | MMD            | C2ST          | KSD           | L2            |
| :---  | :---           | :---          | :---          | :---          |
| SNPE  | 1.2655 ± 0.0412 | 0.9950 ± 0.0000 | 0.0886 ± 0.0292 | 6.6242 ± 0.9629 |
| ASNPE | **1.1216** ± 0.0780 | 0.9950 ± 0.0000 | **0.0633** ± 0.0256 | **3.7100** ± 0.3214 |
**Table 3: Gaussian Mixture**

Note that _L2_ refers to the average $L_2$ distance between $x_o$ (observational data point) and samples $x\sim p(x|\theta), \theta\sim p(\theta|x_o)$. This provides some insight into how well the posterior is calibrated around our data point $x_o$, and whether samples drawn from the posterior $\theta\sim p(\theta|x_o)$ ultimately yield synthetic data close to $x_o$ under the mechanistic model.

While these simulation horizons are relatively small, we find that ASNPE tends to outperform SNPE across most settings and on most metrics. In particular, ASNPE outperforms SNPE on MMD and KSD metrics across all settings. Outside of _Bernoulli GLM_, ASNPE either matches or exceeds SNPE in terms of C2ST and reported $L_2$, whereas on _Bernoulli GLM_ SNPE is better on these metrics.

### Additional studies for our final manuscript

Due to the difficulty of evaluating our transportation simulator for large simulation samples, in-depth ablation studies or hyperparameter sensitivity analysis has been difficult to carry out at scale. Given the efficiency of running our method on the much faster, lower-dimensional simulations explored above, we will use these scenarios to provide 1) hyperparameter ablations, 2) comparison between ASNPE and SNPE over long simulation horizons, and 3) diminishing effects/failure modes of the acquisition function in the SBI loop.

### In defense of our single traffic task

We’d like to note that we intentionally focused our efforts on a single problem that captures many facets of challenging real-world settings, including high-dimensionality, realistic (traffic) dynamics, and slow simulation times. Further, the availability of real-world, metropolitan-scale traffic simulations is very limited, and most high-impact works in the urban mobility space limit their empirical evaluation to synthetic networks or small-scale road networks. This is due to the difficulty of accessing and/or developing realistic road network models for major metropolitan areas. While investigating additional simulation models is undoubtedly valuable and can improve the general appeal of our contribution, we ultimately believe the transportation problem focused on in this paper best captures the realistic qualities of slow, real-world, high-dimensional simulation models.

[1] D. S. Greenberg, M. Nonnenmacher, and J. H. Macke. Automatic posterior transformation for likelihood-free inference, 2019.

[2] Lueckmann, J. M., Boelts, J., Greenberg, D., Goncalves, P., & Macke, J. (2021). Benchmarking simulation-based inference. AISTATS.

---

### Author Response · Authors · 2024-08-07
**Minor correction in table headers**

`NOTE` _Please note the column headers in the global rebuttal tables were accidentally rotated relative to the table bodies, and should instead read_

> | L2            | C2ST          | MMD           | KSD            |
> | :---           | :---          | :---          | :---          |

_Below we reproduce the "Results" section from the global rebuttal with corrected headers and metric references for convenience._

---

### Results

| | L2 | C2ST | MMD | KSD |
| :--- | :--- | :--- | :--- | :--- |
| SNPE | 11.3820 ± 1.1043 | 0.9938 ± 0.0003 | 0.2901 ± 0.1217 | 5.9121 ± 0.3323 |
| ASNPE | **11.2673** ± 0.5796 | 0.9938 ± 0.0002 | **0.2558** ± 0.1331 | **5.6731** ±0.1988 |

**Table 1: SLCP Distractors**

| | L2 | C2ST | MMD | KSD |
| :--- | :--- | :--- | :--- | :--- |
| SNPE | 18.7130 ± 1.8315 | **0.9952** ± 0.0028 | 0.2496 ± 0.0579 | **37.2472** ± 3.4843 |
| ASNPE | **16.1893** ± 0.5625 | 0.9980 ± 0.0007 | **0.2340** ± 0.0467 | 39.4204 ± 0.2686 |

**Table 2: Bernoulli GLM**

| | L2 | C2ST | MMD | KSD |
| :--- | :--- | :--- | :--- | :--- |
| SNPE | 1.2655 ± 0.0412 | 0.9950 ± 0.0000 | 0.0886 ± 0.0292 | 6.6242 ± 0.9629 |
| ASNPE | **1.1216** ± 0.0780 | 0.9950 ± 0.0000 | **0.0633** ± 0.0256 | **3.7100** ± 0.3214 |

**Table 3: Gaussian Mixture**

Note that smaller values are better for each metric, and _L2_ refers to the average $L_2$ distance between $x_o$ (observational data point) and samples $x\sim p(x|\theta), \theta\sim p(\theta|x_o)$. This provides some insight into how well the posterior is calibrated around our data point $x_o$, and whether samples drawn from the posterior $\theta\sim p(\theta|x_o)$ ultimately yield synthetic data close to $x_o$ under the mechanistic model.

While these simulation horizons are relatively small, we find that ASNPE tends to outperform SNPE across most settings and on most metrics. In particular, ASNPE outperforms SNPE on MMD and L2 metrics across all settings. Outside of _Bernoulli GLM_, ASNPE either matches or exceeds SNPE in terms of C2ST and KST, whereas on _Bernoulli GLM_ SNPE achieves better scores for these metrics.

---

### Comment · Area_Chair_VKo6 · 2024-08-08

Dear authors and reviewers,

The authors-reviewers discussion period has now started.

@Reviewers: Please read the authors' response, ask any further questions you may have or at least acknowledge that you have read the response. Consider updating your review and your score when appropriate. Please try to limit borderline cases (scores 4 or 5) to a minimum. Ponder whether the community would benefit from the paper being published, in which case you should lean towards accepting it. If you believe the paper is not ready in its current form or won't be ready after the minor revisions proposed by the authors, then lean towards rejection.

@Authors: Please keep your answers as clear and concise as possible.

The AC

---

### Decision · Program_Chairs · 2024-09-25

**Decision:**

Accept (poster)

**Comment:**

The reviewers recommend acceptance (8-6-4-5-6-7). The paper introduces ASNPE, a new method for simulation-based inference based on active learning. The authors-reviewers discussion has been very constructive and has led to a number of improvements to the paper.  In particular, the reviewers noted that ASNPE was a generic improvement over SNPE and, as such, should be evaluated on common SBI benchmarks instead of focusing only on the task of urban demand calibration. The authors have provided convincing results SBI benchmarks during the rebuttal, which has been largely appreciated by the reviewers. The borderline rejection from Reviewer JwDn is not an obstacle to acceptance, as most the major concerns have been addressed, despite the lack of acknowledgment from the reviewer.

Given the scope of the paper and the potential of the proposed approach beyond the traffic task, the authors should consider the opportunity to reframe the paper and its title to better reflect the generic contribution of ASNPE. Finally, the authors are asked to implement the changes and clarifications discussed with the reviewers in the final version of the paper.